

# Climate tipping point interactions and cascades: A review

Nico Wunderling[1,2,3,*], Anna von der Heydt[4,5,*], Yevgeny Aksenov[6], Stephen Barker[7], Robbin Bastiaansen[4,8], Victor Brovkin[9], Maura Brunetti[10], Victor Couplet[11], Thomas Kleinen[9], Caroline H. Lear[7], Johannes Lohmann[12], Rosa Maria Roman-Cuesta[13], Sacha Sinet[4,5], Didier Swingedouw[14], Ricarda Winkelmann[1,15], Pallavi Anand[16], Jonathan Barichivich[17,18], Sebastian Bathiany[19], Mara Baudena[5,20,21], John T. Bruun[22], Cristiano M. Chiessi[23], Helen K. Coxall[24,25], David Docquier[26], Jonathan Donges[1,2,3], Swinda K.J. Falkena[4], Ann Kristin Klose[1,15], David Obura[27], Juan Rocha[2], Stefanie Rynders[6], Norman Julius Steinert[28], and Matteo Willeit[1]

[1]Potsdam Institute for Climate Impact Research (PIK), Member of the Leibniz Association, Potsdam, Germany
[2]Stockholm Resilience Centre, Stockholm University, Stockholm, Sweden
[3]High Meadows Environmental Institute, Princeton University, Princeton, USA
[4]Institute for Marine and Atmospheric research Utrecht (IMAU), Department of Physics, Utrecht University, Utrecht, The Netherlands
[5]Centre for Complex Systems Studies, Utrecht University, Utrecht, the Netherlands
[6]National Oceanography Centre, Southampton, United Kingdom
[7]School of Earth and Environmental Sciences, Cardiff University, United Kingdom
[8]Department of Mathematics, Utrecht University, Utrecht, The Netherlands
[9]Max Planck Institute for Meteorology, Hamburg, Germany
[10]Group of Applied Physics and Institute for Environmental Sciences, University of Geneva, Geneva, Switzerland
[11]Earth and Life Institute, UCLouvain, Louvain-la-Neuve, Belgium
[12]Physics of Ice, Climate and Earth, Niels Bohr Institute, University of Copenhagen, Denmark
[13]European Commission, Joint Research Center, Sustainable Resources, Forests and Bioeconomy Unit., Ispra, Italy
[14]Environnements et Paléoenvironnements Océaniques et Continentaux (EPOC), Univ. Bordeaux, CNRS, Bordeaux INP, France
[15]Institute of Physics and Astronomy, University of Potsdam, Potsdam, Germany
[16]School of Environment, Earth and Ecosystem Sciences, The Open University, Milton Keynes, United Kingdom
[17]Laboratoire des Sciences du Climat et de l'Environnement (LSCE), LSCE/ IPSL, CEA-CNRS-UVSQ, Université Paris-Saclay, Gif-sur-Yvette, France
[18]Instituto de Geografı́a, Pontificia Universidad Catoĺica de Valparaı́so, Valparaı́so, Chile
[19]Technical University Munich, Munich, Germany
[20]Institute of Atmospheric Sciences and Climate, National Research Council of Italy (CNR-ISAC), Torino, Italy
[21]National Biodiversity Future Center, Palermo, Italy
[22]Faculty of Environment, Science and Economy, University of Exeter
[23]School of Arts, Sciences and Humanities, University of São Paulo, São Paulo, Brazil
[24]Department of Geological Science, Stockholm University, Stockholm, Sweden
[25]The Bolin Centre for Climate Research, Stockholm, Sweden
[26]Royal Meteorological Institute of Belgium, Brussels, Belgium
[27]CORDIO East Africa, Mombasa, Kenya
[28]NORCE Norwegian Research Centre, Bjerknes Centre for Climate Research, Bergen, Norway
[*]These authors contributed equally to this work.

**Correspondence:** Nico Wunderling (nico.wunderling@pik-potsdam.de), Anna von der Heydt (a.s.vonderheydt@uu.nl)





**Abstract.** Climate tipping elements are large-scale subsystems of the Earth that may transgress critical thresholds (tipping points) under ongoing global warming, with substantial impacts on biosphere and human societies. Frequently studied examples of such tipping elements include the Greenland Ice Sheet, the Atlantic Meridional Overturning Circulation, permafrost, monsoon systems, and the Amazon rainforest. While recent scientific efforts have improved our knowledge about individual
5   tipping elements, the interactions between them are less well understood. Also, the potential of individual tipping events to induce additional tipping elsewhere, or stabilize other tipping elements is largely unknown. Here, we map out the current state of the literature on the interactions between climate tipping elements and review the influences between them. To do so, we gathered evidence from model simulations, observations and conceptual understanding, as well as archetypal examples of paleoclimate reconstructions where multi-component or spatially propagating transitions were potentially at play. Lastly, we
10   identify crucial knowledge gaps in tipping element interactions and outline how future research could address those gaps.

# 1 Introduction

## 1.1 Climate tipping elements

Climate change can cause abrupt and irreversible environmental and societal change (Masson-Delmotte et al., 2021). Several climate subsystems have been identified as at risk of undergoing qualitative and often irreversible change when critical thresholds of global warming are transgressed (Wang et al., 2023; Armstrong McKay et al., 2022; Bathiany et al., 2016; Lenton et al., 2008). Such subsystems are termed tipping elements (TEs) and examples include the Atlantic Meridional Overturning Circulation (AMOC), polar ice sheets, tropical rainforests, permafrost regions and the marine biosphere. Nonlinear changes can occur at tipping points (TPs), where a slight change in a parameter or a small perturbation of a system's state can cause a large change in the system, driving it to transit into a completely different (often undesirable) state. From a dynamical systems point of view, a tipping point can be reached when passing a critical value of a control parameter, for example, the atmospheric $CO_2$ concentration, which affects the equilibrium states of the system. These processes are at the heart of tipping behavior in the climate system and were found in numerous subsystems of the climate system.

In the context of this paper, we refer to a tipping element as any climate subsystem that has a nonlinear response (self-amplifying feedback) to forcing so that the system reorganizes (Armstrong McKay et al., 2022). This definition includes large-scale climate tipping elements such as the AMOC (Weijer et al., 2019) or polar ice sheets (Rosier et al., 2021), where the associated feedbacks (e.g. salt-advection, melt-elevation, or ice-albedo) are well known, but also more regional bistabilities between savanna and forest vegetation in the Amazon region. In addition, we also consider elements that can show nonlinear behavior without being tipping elements on their own.

Tipping processes involving several tipping elements can also be found in Earth's history: during the last ice age, repeated abrupt shifts, so-called Dansgaard-Oeschger events, occurred between cold and warm phases lasting 1,000–4,000 years (Dansgaard et al., 1993). While mostly polar and Northern Hemisphere elements (sea ice, ocean circulation, atmospheric dynamics) appear to have been involved (Vettoretti and Peltier, 2016; Zhang et al., 2014; Clement and Peterson, 2008; Ganopolski and Rahmstorf, 2001), the climate impact of these shifts was global (Barbante et al., 2006; Shackleton et al., 2000).





## 1.2 Interactions in the Earth's climate system

Most climate subsystems are linked via circulation systems in the ocean and atmosphere, which leads to statistical associations between them in their natural variability, often called teleconnections. For example, El Niño-Southern Oscillation (ENSO), the monsoon systems and Atlantic multidecadal variability form global modes of climate variability (Kravtsov et al., 2018; Dommenget and Latif, 2008). In addition, sea surface temperature variability in the North Pacific coupled to tropical variability (ENSO) is transferred to other regions via atmospheric teleconnections and amplified on longer time scales by the large ocean

heat capacity (Dommenget and Latif, 2008). Similarly, multidecadal variability originating in the North Atlantic Ocean (Knight et al., 2005; Delworth and Mann, 2000), which is believed to be partly connected to the AMOC (Buckley and Marshall, 2016), has a global expression in sea surface temperature patterns due to the interaction of slow oceanic and fast (but large-scale) atmospheric processes (Kravtsov et al., 2018). Hence, most nonlinear climate subcomponents are not isolated from each other, but are connected either directly or mediated via changes to the background state (Liu et al., 2023; Kriegler et al., 2009). Via

such connections (see Fig. 1) tipping in one subsystem – the leading subsystem – can therefore cause tipping in another one – the following subsystem (Klose et al., 2020; Dekker et al., 2018). Here we call the linkages between tipping elements and/or other nonlinear components tipping interactions, whether they have a stabilizing or a destabilizing effect. The most extreme case is the situation in which the tipping of element A causes a subsequent tipping of element B. In this paper, we define a sequence of events involving several nonlinear components of the Earth system as tipping cascades (Dekker et al., 2018;

Wunderling et al., 2021a). These tipping cascades can come in various forms dependent on the ordering of tipping elements (e.g. Klose et al., 2021) and might involve different mathematical bifurcations such as fold- and Hopf-bifurcations (e.g. Dekker et al., 2018). Eventually a tipping cascade might result in a fundamental change in the Earth's equilibrium climate.

For example, an abrupt change in AMOC strength can lead to an intensification of ENSO, while a disintegration of the Greenland Ice sheet can lead to an abrupt AMOC shift. We do not restrict our definition to specific spatial scales, time scales

or severity of impact of the tipping elements. Therefore, also the slow local invasion fronts in spatial (eco)systems would be considered (e.g. Bel et al., 2012). Interactions between climate tipping elements could effectively lower the thresholds for triggering a tipping event or cascade as compared to individual tipping elements (Wunderling et al., 2021a; Klose et al., 2020). Moreover, a tipping cascade could activate processes leading to additional $CO_2$ emissions into the atmosphere; permafrost thaw and forest dieback are typical examples of such feedbacks (Wunderling et al., 2020; Lenton et al., 2019; Steffen et al.,

60 2018).

Due to the many nonlinearities in the climate system, it is also conceivable that components of the Earth system, though not necessarily tipping elements on their own, could mediate or amplify nonlinear transitions in one component, creating larger-scale impacts also in other components. A prominent example is Arctic summer sea ice cover, which shows an almost linear response to the $CO_2$ forcing and is not expected to show tipping behavior under anthropogenic forcing (Lee et al., 2021),

nevertheless can still sharpen and amplify transitions in the ocean-atmosphere-cryosphere system (Gildor and Tziperman, 2003). On the other hand, an abrupt transition in one tipping element may also stabilize other climate subsystems (e.g. Nian





et al., 31 May 2023; Sinet et al., 2023) as is the case for a weakening AMOC decreasing local temperatures around Greenland (Jackson et al., 2015).

### 1.3 Motivation and structure of this work

While most TEs that have been proposed so far are clearly regional (with some being large scale), there are significant knowledge gaps with respect to their tipping probability, impact estimates, time scales, as well as their interactions. The potential of a tipping cascade that could lead to a global reorganization of the climate system (Steffen et al., 2018; Hughes et al., 2013) remains therefore speculative. However, since multiple individual tipping point thresholds may be crossed during this century with ongoing global warming, and lead to severe tipping element interactions and cascading transitions in the worst case, it is
critical to review the current state of the science and reveal research gaps that need to be filled in (Armstrong McKay et al., 2022; Masson-Delmotte et al., 2021; Rocha et al., 2018).

Here we provide an overview of the current knowledge of tipping element interactions and the potential for tipping cascades. Even though most potential tipping elements are regional, it does not necessarily take a tipping cascade in order to obtain global climate effects. Examples for such globalizing effects are sea level rise, and the emission of greenhouse gases. Moreover, as
mentioned above, a "cascade" does not necessarily involve a sequence of elements that all have individual tipping points, but can also arise from dynamically stable but still nonlinear elements (such as sea ice switches). We therefore do not restrict ourselves to the most plausible tipping elements in this review, but also include nonlinear components like Arctic sea ice, ENSO and monsoon systems that can act as mediators of tipping events in the Earth system.

The main part of this paper reviews the current knowledge of interactions between specific pairs of tipping components
(section 2). In section 3, we discuss three archetypal paleoclimate candidates of tipping sequences that involve more than one tipping element: one from the more distant past (Eocene-Oligocene transition; approximately 34 million years ago), one from the more recent past (Dansgaard-Oeschger events, Bølling–Allerød warm period, and Heinrich events; during and since the last glacial period), and a paleoclimatic perspective on interactions between AMOC and the Amazon rainforest. Further, we discuss a contemporary and illustrative example of a cascade between parts of tipping elements, where the decline of Arctic
sea ice deteriorates coastal permafrost through increased erosion (section 4). Next, we map out the present state of modeling tipping sequences with respect to the role of complex Earth system models and more conceptualized approaches (section 5). Lastly, we discuss current research gaps and ways forward from a knowledge, a modeling and a data perspective. We also discuss the value of newly arising methods from machine learning and Earth-observation, and how they could complement the present research on interacting climate tipping elements. Finally, we conclude the recent progress on tipping cascades and
interactions between tipping elements (both section 6).



## 2 Interactions between climate tipping elements and nonlinear climate components

### 2.1 Interactions across scales in space and time

In this section, we lay out the current state of the literature on the interaction processes between components that are known to show nonlinear behavior or are even suspected tipping elements. The summary of the detailed sections 2.2–2.8 is shown
in Table 1 and Figures 1 & 2. We summarize that these elements are not isolated entities but interact across the entire globe (Fig. 1). Not only do the interactions span global distances, but the elements themselves are systems of sub-continental up to (nearly) global spatial scale that may tip on temporal scales of months up to millennia, i.e. tipping elements interact across scales in space and time (Fig. 2) (Rocha et al., 2018; Kriegler et al., 2009). The respective processes of the interactions can be found in Table 1, alongside an estimation of the interaction direction and, if available, an estimation of their strength (based on
the in-detail literature review of sections 2.2–2.8).

Some tipping elements are of sub-continental spatial scale (e.g., coral reefs or the Greenland Ice Sheet), while others cover significant portions of the globe (e.g., AMOC). Also the temporal scales differ vastly among the different climate tipping elements: some of them are considered fast tipping elements once a tipping process has been initiated (tipping on the order of (months) years/decades to centuries, e.g., Amazon rainforest and AMOC), while others are considered slow tipping elements
(tipping on the order of centuries to millennia, e.g. Greenland Ice Sheet). These individual dynamics on space and time of the individual tipping elements are therefore also important for their interactions (mapped out in Fig. 2).

### 2.2 Interactions between ice sheets and AMOC

The AMOC, Greenland Ice Sheet (GIS) and West Antarctic Ice Sheet (WAIS) are core tipping elements and are threatened by
increasing $CO_2$ emissions (Armstrong McKay et al., 2022; Pörtner et al., 2019). Moreover, GIS, AMOC, and WAIS interact on very different timescales ranging from decades to multiple centuries. While some of those links might be stabilizing, others are destabilizing and would allow for the possibility of large-scale cascading events.

#### 2.2.1 Differing North Atlantic from Ocean meltwater effects

Greenland Ice Sheet to AMOC: The AMOC depends on the formation of dense water in the high latitudes of the North At-
lantic. In its present state, this process is widely sustained by the positive salt-advection feedback (Weijer et al., 2019) – as the AMOC transports salt northward, a higher surface water density is maintained in this region. As GIS melting increases, the associated discharge of freshwater in the ocean would result in a decrease of the surface water density, inhibiting the formation of dense waters through deep convection and thereby weakening the circulation. As less salt is transported to the North Atlantic, the salt-advection feedback implies a self-sustained freshening of the high latitudes of the North Atlantic, which, in
the worst case, can result in the collapse of the AMOC. On top of this classical positive feedback, there exists a wide range of other feedbacks related to the AMOC, either negative (heat advection feedback, e.g. Swingedouw et al. (2007)) or positive





**Table 1.** List of links between elements (tipping elements and other nonlinear components) discussed in this paper. We list directed links with a short summary of the main physical mechanism(s) underlying the specific connection. Further, an estimate of whether the effect is stabilizing or destabilizing the impacted element is given. If there is incipient evidence in the literature, we also add an indication of the qualitative strength of the response (weak, moderate, strong). Otherwise the link and/or its strength is set as unclear. Lastly, key references for the specific link are noted. Abbreviations: AABW = Antarctic Bottom Water, AMAZ = Amazon rainforest, AMOC = Atlantic Meridional Overturning Circulation, ENSO = El Niño-Southern Oscillation, GIS = Greenland Ice Sheet, ISM = Indian Summer Monsoon, PERM = Permafrost, SST = Sea Surface Temperature, WAIS = West Antarctic Ice Sheet, WAM = West African Monsoon.

| Link between elements | Physical processes | Response (destabilizing, stabilizing, unclear) [Strength] | Key references |
|---|---|---|---|
| GIS ⟶ AMOC (Section 2.2.1) | Freshwater influx from GIS into the North Atlantic weakens the AMOC (consistent across models), uncertain how much freshwater flux is necessary to shut down AMOC | Destabilizing [Strong] | Weijer et al. (2019); Mecking et al. (2016); Stouffer et al. (2007) |
| WAIS ⟶ AMOC (Section 2.2.1) | Competing effects: (i) Freshwater from the Southern Ocean and associated AABW reduction imply a strengthening of the AMOC (deep ocean adjustment, likely on short timescales (years to decades) through wave dynamics, (ii) Freshwater from the Southern Ocean reaches North Atlantic and weakens the AMOC (likely on timescales from decades to centuries), (iii) Increasing wind intensity over Southern Hemisphere and sea ice cover over Southern Ocean can strengthen the AMOC | Unclear, likely timescale dependent [Weak/Moderate] | Swingedouw et al. (2009, 2008); Stouffer et al. (2007); Seidov et al. (2005) |
| AMOC ⟶ GIS (Section 2.2.2) | Reduced northward heat transport in the Atlantic implies substantial cooling of the Northern Hemisphere | Stabilizing [Strong] | Jackson et al. (2015); Stouffer et al. (2006) |
| AMOC ⟶ WAIS (Section 2.2.2) | Reduced northward heat transport in the Atlantic may lead to warming of the Southern Ocean, destabilizing WAIS | Destabilizing [Unclear] | Bintanja et al. (2013); Swingedouw et al. (2008) |
| GIS ⟶ WAIS (Section 2.2.3) | Sea level rise through GIS melt destabilizes WAIS, potentially significantly since most of the WAIS bedrock lies below sea level (marine ice sheet instability) | Destabilizing [Moderate] | Gomez et al. (2020); Kopp et al. (2010) |
| WAIS ⟶ GIS (Section 2.2.3) | Sea level rise through WAIS melt destabilizes GIS, potentially weakly since most of the GIS bedrock lies above sea level | Destabilizing [Weak] | Gomez et al. (2020); Mitrovica et al. (2009) |
| Arctic sea ice ⟶ AMOC (Section 2.3) | Studies modeling contemporary climate find that AMOC weakens as a result of Arctic sea ice decline due to southward advection of lighter water to the North Atlantic convection sites (reasons: (1) strong anomalous heat flux into the ocean due to lowering albedo; and (2) a small and transient contribution of freshwater influx from melting Arctic sea ice) | Destabilizing [Weak/Moderate] | Liu and Fedorov (2022); Li et al. (2021); Jansen et al. (2020); Sévellec et al. (2017) |
| AMOC ⟶ Arctic sea ice (Section 2.3) | A weaker AMOC results in lower Atlantic Ocean heat transport and slows the pace of the current Arctic sea ice loss | Stabilizing [Weak/Moderate] | Liu and Fedorov (2022); Liu et al. (2020); Delworth et al. (2016) |



| Link between elements | Physical processes | Response (destabilizing, stabilizing, unclear) [Strength] | Key references |
|---|---|---|---|
| AMOC ⟶ AMAZ (Section 2.4) | A collapsing AMOC would reduce (increase) SSTs in the North (South) Atlantic and cause a southward shift of the tropical rain belt. Therefore, precipitation would decrease in the northern part of the Amazon and increase in the southern part. However, exact locations of precipitation changes are model dependent | Unclear [Moderate] | Bellomo et al. (2023); Jackson et al. (2015); Parsons et al. (2014) |
| AMOC ⟶ ENSO (Section 2.5.1) | A weakening AMOC increases the equator-to-pole temperature gradient and may therefore strengthen the trade winds in general. Feedbacks generating ENSO depend on the Pacific background climate, i.e. on trade wind strength. Therefore, ENSO might be found more often in a positive condition | Destabilizing [Weak] | Orihuela-Pinto et al. (2022b); Dekker et al. (2018); Timmermann et al. (2007) |
| ENSO ⟶ AMOC (Section 2.5.1) | Enhanced El Niño can decrease Atlantic hurricane intensity and so limit wind induced AMOC forcing | Unclear [Unclear] | Kim et al. (2023); Ayarzagüena et al. (2018) |
| ENSO ⟶ AMAZ (Section 2.5.2) | Changes in ENSO amplitude and frequency will affect rainfall and temperature variability (including extreme conditions such as droughts) in various regions of the South American continent. This may regionally lead to rainforest dieback and transition to degraded savanna-like ecosystems. Regions in the Andes and western Amazonia seem less vulnerable | Destabilizing [strong, regionally different] | Duque-Villegas et al. (2019); Jiménez-Muñoz et al. (2016) |
| ENSO ⟶ WAIS (Section 2.5.3) | More frequent (or stronger) El Niño events cause more frequent atmospheric blocking situations, leading to West Antarctic ice melt events. This is followed by warm marine air anomalies over West Antarctica, and especially over Ross and Amundsen Sea Embayment regions | Destabilizing [unclear, probably weak/moderate] | Scott et al. (2019); Paolo et al. (2018); Nicolas et al. (2017) |
| ENSO ⟶ Coral Reefs (Section 2.5.4) | ENSO drives abnormally high ocean temperatures, which are superimposed on already warming oceans, leading to severe regional to global bleaching events. With increased warming, bleaching is decoupling from warm ENSO phases, becoming increasingly frequent during cold ENSO phases (La Niña), and at quasi annual frequencies at many locations. At thresholds of 1.5 and 2°C warming, the expected die-off percentage is 70-90% and 90-99%, respectively | Destabilizing [Strong] | Muñiz-Castillo et al. (2019); Lough et al. (2018); Veron et al. (2009) |
| AMOC/ENSO ⟶ Monsoon systems (Section 2.6) | An AMOC weakening would cause a southward shift of the Intertropical convergence zone and the tropical rain belt (e.g., WAM and ISM). ENSO shifts in a warmer climate could lead to an opposing effect on its relationships with the monsoon systems, for example, the ISM-ENSO relationship becomes weaker while the relationship between WAM-ENSO becomes stronger with the warming climate | Highly uncertain: Unclear (monsoon system dependent) [unclear] | Orihuela-Pinto et al. (2022a); Wassenburg et al. (2021); Swingedouw et al. (2009) |
| PERM ⟶ hydrological cycle ⟶ AMOC (Section 2.7) | Under global warming, the PERM region could become wetter and lakes could emerge. If these lakes drain out, the global hydrological cycle becomes weaker. Then, the AMOC weakens | Highly uncertain: destabilizing for AMOC [unclear] | de Vrese et al. (2023); Nitzbon et al. (2020) |



**Figure 1.** Interactions between tipping elements on a world map. All tipping elements discussed in this review article are shown together with their potential connections. The causal interactions links can have stabilizing (blue), destabilizing (red), or unclear (gray) effects. For some elements, it is speculative whether they are tipping elements on their own (such as ENSO or the Arctic sea ice) and they are denoted as such (blue outer ring) but they are included if they play an important role in mediating transitions towards (or from) core tipping elements (compare Section 1.2). Tipping elements that exert a notable feedback on global mean temperature when they tip are denoted by a red inner ring. This temperature feedback can be positive (i.e. amplifying warming, as likely for the permafrost, the Arctic sea ice, the Greenland and West Antarctic ice sheets, the Amazon rainforest and ENSO) or negative (i.e. dampening warming, as likely for the AMOC).



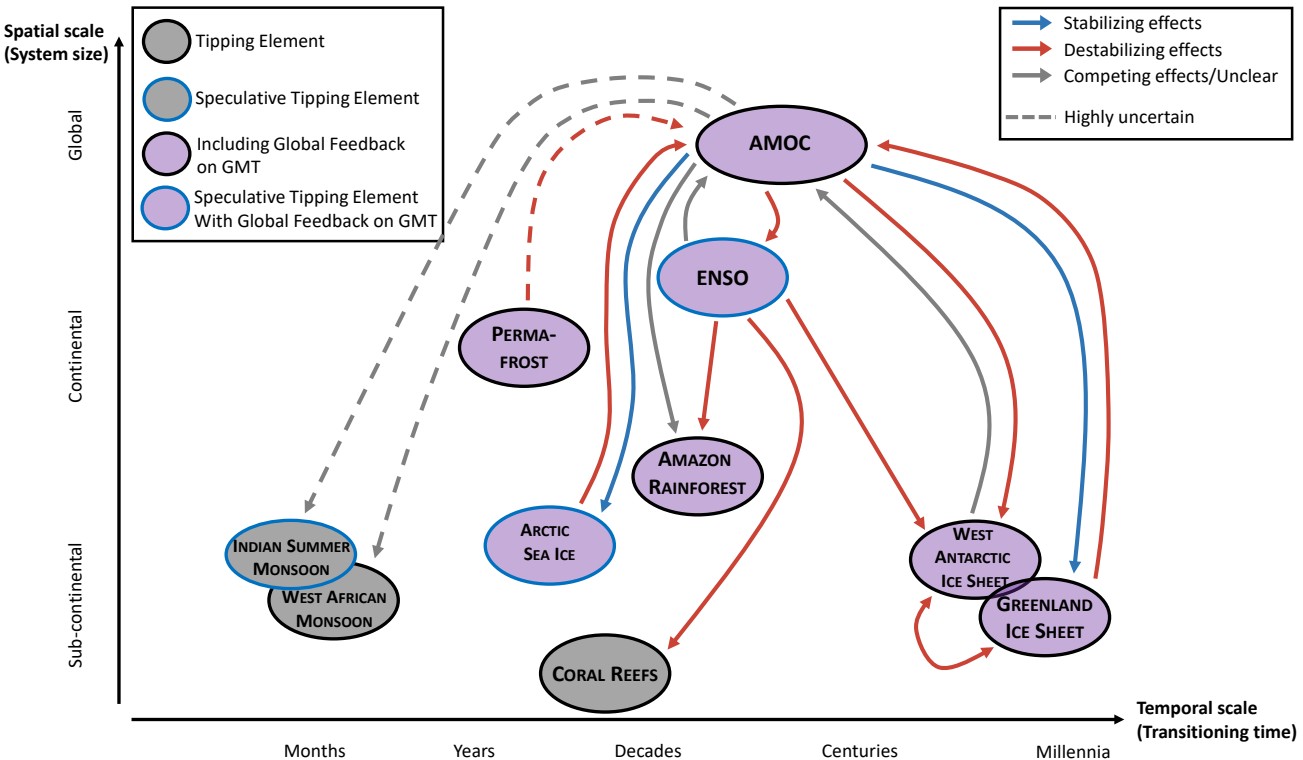

**Figure 2.** Interactions between tipping elements across scales in space and time. Temporal scales are transitioning times of a disintegrating tipping element from months up to millennia. Spatial scales denote the system size from sub-continental to (nearly) global scales. Transitioning times are taken from Armstrong McKay et al. (2022), and spatial scales from Winkelmann et al. (2022). The causal links can be stabilizing (blue), destabilizing (red), or unclear (gray). Some tipping elements are particularly speculative (such as ENSO or the Arctic sea ice) and denoted as such (blue border). Tipping elements that exert a feedback on the global mean temperature when they tip are shown with purple shading.

(evaporation feedback). An overall destabilizing impact of GIS melting on the AMOC is mostly consistent across models, where adding freshwater in the North Atlantic (e.g. Jackson and Wood, 2018; Mecking et al., 2016; Stouffer et al., 2007), also in combination with increasing $CO_2$ emissions (Bakker et al., 2016; Swingedouw et al., 2006), leads to a substantial weakening

of the circulation. Importantly, in the case of a collapse of the AMOC, some models suggest that the AMOC does not recover within human timescale (Jackson and Wood, 2018; Mecking et al., 2016). At the moment, one of the key limitations relating GIS melting and the respective AMOC response concern the way meltwater is spread along Greenland towards the open ocean. This lateral diffusion is mainly performed by oceanic eddies, whose spatial scale is of the order of 10 km at those latitudes, and necessitate oceanic resolution in AOGCMs (Atmosphere/Ocean General Circulation Model) of the order of 2-3 km to be

properly resolved. As a consequence, the spread of GIS meltwater towards the convection sites in the Labrador, Irminger and Nordic Seas, might be underestimated in AOGCMs, which might strongly diminish the potential impact of GIS melting on





the AMOC (Martin and Biastoch, 2023; Swingedouw et al., 2022). Thus, while there exist a few attempts to couple AOGCMs with ice sheet models (e.g. Madsen et al., 2022; Kreuzer et al., 2021; Ackermann et al., 2020; Muntjewerf et al., 2020), the simulated impact of GIS melting on the AMOC remains moderate, but might be underestimated.

West Antarctic Ice Sheet to AMOC: In the case of a freshwater release in the Southern Hemisphere originating from West Antarctica, different opposing processes are at play that could affect the AMOC. These effects have been identified to act on different timescales and depend on the state of the circulation (Berk et al., 2021; Swingedouw et al., 2009). First, the weakening of AABW formation might lead to enhancement of the AMOC through the so-called ocean bipolar seesaw related to deep ocean adjustment through oceanic large-scale waves. Second, the increase in wind intensity over the Southern Hemisphere, related

to an increase in sea-ice cover (Li et al., 2023; Swingedouw et al., 2008), might also help to enhance the AMOC. Third, if large enough, the release of freshwater in the Southern Ocean might eventually reach the North Atlantic on a longer timescale (centuries), possibly weakening the AMOC. As a result, the impact of a WAIS collapse on the AMOC is still unclear, as most models show either a slight weakening (e.g. Stouffer et al., 2007; Seidov et al., 2005) or a slight strengthening(e.g. Swingedouw et al., 2009) of the circulation. Notably, some studies also found that a sufficient freshwater release into the Southern Ocean

allows for delaying an AMOC collapse (Sadai et al., 2020), recovering from it (Weaver et al., 2003) or even avoiding it (Sinet et al., 2023). In most cases, the impact of the WAIS melting on the AMOC remains moderate and mainly affects the southern part of the AMOC.

### 2.2.2   Effects of a collapsing AMOC on ice sheets

An AMOC collapse would imply a decreased northward heat transport, leading to a substantial cooling of the Northern Hemi-
sphere, along with warming in the Southern Hemisphere (Pedro et al., 2018; Jackson et al., 2015; Stouffer et al., 2006). Cooling the high latitudes of the North Atlantic would stabilize the GIS, possibly allowing for a safe overshoot of the GIS tipping point (Wunderling et al., 2023; Ritchie et al., 2021). Conversely, the related warming of the Southern Ocean represents a destabilizing impact on the WAIS, being susceptible to these warmer ocean waters via the ice shelves and their buttressing effect on upstream ice flow (Favier et al., 2014; Joughin et al., 2014).

### 2.2.3   Direct interactions between Greenland and West Antarctic Ice Sheets via sea level feedbacks

It is known that an increase in sea level has an overall destabilizing influence on marine-based sectors of ice sheets, possibly triggering or enhancing the retreat of their grounding line (Schoof, 2007; Weertman, 1974). In the case of an ice sheet collapse, the induced sea level rise from ice-sheets would vary locally depending on gravitational effects, rotational effects and mantle deformation (Kopp et al., 2010; Mitrovica et al., 2009). Overall, sea level rise is expected to negatively impact both the GIS
and WAIS, but more strongly the latter where most of the bedrock lies well below sea level (Gomez et al., 2020).



## 2.3  Interactions between AMOC and Arctic sea ice

The strength of the AMOC is controlled by the deep convective activity at different sites in the North Atlantic Ocean (Labrador, Irminger and Nordic Seas), which is largely driven by its high surface density (Kuhlbrodt et al., 2007). Changing Arctic sea ice cover can modulate the latter, and thus the AMOC, mainly in two ways (Sévellec et al., 2017).

First, it alters radiative heating and ocean-atmosphere heat loss via changing albedo. More precisely, as the Arctic sea ice area has substantially decreased in the past 40 years, especially during summer months (Masson-Delmotte et al., 2021), the open water fraction of the Arctic Ocean has increased and will continue to do so in the future (Crawford et al., 2021). This has led to an increase in the absorption of solar radiation and to subsequent ocean warming, which can propagate to convection areas.

Second, changes in Arctic sea ice alter the ocean density by brine rejection during sea ice formation, or conversely by freshening from sea ice melt. In particular, the recent decrease in Arctic sea ice area, together with the ice loss from the Greenland Ice Sheet (Section 2.2.1), has added freshwater to the Arctic Ocean, although the trend in freshwater content has slowed down during the past decade (Solomon et al., 2021). According to model simulations performed by Sévellec et al. (2017), these warm and fresh anomalies coming from sea ice melting, could propagate southward to the subpolar North Atlantic

Ocean. These would affect the deep ocean at the main convection sites by reducing the surface density and would thus weaken the AMOC. The estimated timescale of this propagation is multi-decadal (Liu and Fedorov, 2022; Li et al., 2021; Sévellec et al., 2017).

The AMOC can also affect Arctic sea ice via the transport of warm water to the North Atlantic Ocean, and subsequently to the Arctic Ocean via the Barents Sea Opening and Fram Strait. A weaker AMOC could result in lower ocean heat transport

and increased Arctic sea ice area (Delworth et al., 2016). The estimated timescale of this effect is approximately one year (Liu and Fedorov, 2022). However, recent observations show that the ocean heat transport to the Arctic has increased, especially on the Atlantic side (Docquier and Koenigk, 2021; Polyakov et al., 2017; Onarheim et al., 2015; Årthun et al., 2012). Thus, the effect of a decreasing AMOC may merely slow down the pace of ongoing increases in ocean heat transport and the associated decrease in Arctic sea ice (Liu et al., 2020).

Despite this tight link between the AMOC and Arctic sea ice, Arctic summer sea ice cannot be considered a tipping element. A tipping point in the Arctic Ocean would mean that it loses so much sea ice that the reduced albedo results in enough warming to prevent sea ice from forming again once melted. However, according to model simulations in which a summer ice-free Arctic Ocean is simulated, Arctic sea ice recovers within two years, suggesting that the ice-albedo feedback is alleviated by large-scale recovery mechanisms (Tietsche et al., 2011). Winter sea ice extent does show very different states at the same atmospheric $CO_2$

concentration, attributed to AMOC strength (Schwinger et al., 2022).





## 2.4 Effects of AMOC changes on the Amazon rainforest

The strength of the AMOC exerts a substantial influence on the climate of tropical South America, most importantly, on rainfall and its seasonal distribution. This in turn affects the state and stability of another potential tipping element in the Earth system: The Amazon rainforest.

The most important large-scale effect of the AMOC on rainfall in the Amazon works via the pattern of SSTs in the Atlantic, and the associated shifts in the ITCZ (Intertropical Convergence Zone) and the tropical rain belt. There is widespread agreement that a reduction or even collapse of the AMOC would lead to reduced SSTs in the North Atlantic and increased SSTs in the South Atlantic (Bellomo et al., 2023; Manabe and Stouffer, 1995). This change is caused by the reduction in the AMOC-related northward ocean heat transport and is amplified by wind-evaporation-feedbacks (Orihuela-Pinto et al., 2022a). The changed

SST pattern in turn affects atmospheric circulation by strengthening the Northern Hemisphere Hadley cell particularly during boreal winter (Bellomo et al., 2023). As the location of the tropical rain belt depends on the cross-equatorial energy flux and the atmosphere energy input close to the equator (Bischoff and Schneider, 2014; Schneider et al., 2014), a weakened AMOC together with a persistent Southern Ocean warming lead to a southward migration of the tropical rain belt, depending on the $CO_2$-forcing trajectory (Kug et al., 2022). Hence, AMOC-weakening may cause a tropical rain belt shift. This southward shift

would cause a substantial reduction in rainfall over northern South America, and an increase in rainfall over the portion of the Amazon located in the Southern Hemisphere, as well as over northeastern Brazil which is directly affected by the tropical rain belt (Jackson et al., 2015). Nevertheless, over the Amazon basin, the extent of this migration is model dependent (e.g. Swingedouw et al., 2013; Stouffer et al., 2006). Indeed, while the northern part of the Amazon might experience a decrease in precipitation, the southern part, on the opposite, might see enhanced precipitation, which has the potential to stabilize the

rainforest there (Ciemer et al., 2021). The limit between the two regions is where model dependency is strongest, resulting in a large uncertainty concerning the potential impact of AMOC weakening in the Amazon rainforest dieback.

    To conclude, although different Earth system models have different biases in the location, shape and strength of the tropical rain belt, they generally agree on the AMOC-collapse induced increase in precipitation over the southern portion of the Amazon and northeastern Brazil (Bellomo et al., 2023; Nian et al., 31 May 2023; Orihuela-Pinto et al., 2022a; Liu et al., 2020). Given

that the forests in the southern half of the basin contribute mostly to the rainfall generation over the basin (Staal et al., 2018), one could speculate that this would lead to a stabilization of the Amazon, given that a substantial fraction (24-70%, Baudena et al. (2021) and references therein) of the rainfall of the basin is nonetheless produced by local moisture recycling. Furthermore, it has been shown that the altered tropical rain belt dynamics throughout the year would mean a reduction of rainfall mostly during the current wet season (peaking around March), and an increase in the dry season, peaking around September (Campos

et al., 2019; Parsons et al., 2014). Importantly, the consequences for the rainforest of a more equalized annual cycle are unclear. More generally, the full spectrum of rainforest stressors including societal-driven pressures, such as land-use changes driving deforestation, has to be taken into account when assessing AMOC effects over the Amazon rainforest (Lovejoy and Nobre, 2018).



## 2.5 Influences of ENSO on proposed tipping elements

The El Niño-Southern Oscillation (ENSO) is the most important mode of climate variability on interannual time scales, fundamentally affecting regional and global atmospheric and oceanic circulation (McPhaden et al., 2006). The response to climate change of ENSO itself still remains debated, mainly because there are multiple (positive and negative) feedback processes in the tropical Pacific ocean-atmosphere system, whose relative strength determines the response of ENSO variability (Timmermann et al., 2018; Cai et al., 2015). Further, recent studies disagree about the future frequency of El Niño phases under global

warming (Cai et al., 2021; Wengel et al., 2021). In particular, a decreasing frequency of El Niño phases under global warming was suggested by a global climate model resolving meso-scale oceanic eddies and consequently reduced biases in the tropical oceanic mean state (Wengel et al., 2021). Although it is debated or even unlikely whether ENSO should be considered a tipping element in itself (Armstrong McKay et al., 2022), it exerts important feedbacks on other global tipping elements. Through its global teleconnections, ENSO has the potential to influence multiple Earth system components including the AMOC, the

Amazon rainforest and the West Antarctic Ice Sheet. Changes in ENSO amplitude or frequency could alter the strength of (stabilizing or destabilizing) feedbacks within other (remote) tipping elements. Therefore, in this section, we discuss possible interactions between ENSO and other tipping elements.

### 2.5.1 Interactions between ENSO and AMOC

Various physical mechanisms have been discussed to explain how a decline in strength or complete shutdown of the AMOC

could affect ENSO variability, mostly in terms of the amplitude of ENSO. An AMOC decline typically leads to cooling in North Atlantic surface temperatures, which affects the global atmospheric circulation, including the trade winds in the tropical Pacific. In many GCMs, upon decline of the AMOC, the northeasterly trade winds are intensified and the Intertropical Convergence Zone (ITCZ) displaces southwards, eventually leading to an intensification of ENSO amplitude through nonlinear interactions (Timmermann et al., 2007). While the response of the trade winds and ITCZ to AMOC decline seems to be relatively robust

within different (generations of) GCMs, the response in ENSO magnitude or frequency is much more model dependent: Trade winds can also affect the thermocline depth in the eastern tropical Pacific thereby weakening the ENSO (Timmermann et al., 2007). By that, the zonal structure of winds and stratification is affected leading to zonal shifts of variability patterns but no significant change in amplitude (Williamson et al., 2018). Alternatively, weaker air-sea coupling due to altered trade winds affects the relevant tropical Pacific feedback balance such that the growth rate of ENSO is significantly reduced (Orihuela-

Pinto et al., 2022b). In another model study, where physically based, conceptual models of the AMOC (box model) and ENSO (Zebiak-Cane model, Zebiak and Cane (1987)) are coupled via the trade wind strength it was found that an AMOC collapse intensifies the tropical Pacific trade winds and shifts the ENSO system further into its oscillatory mode (i.e., amplitude increase) (Dekker et al., 2018). It should be noted that most GCMs still exhibit severe biases in tropical temperature patterns, partly caused by not properly resolved oceanic mesoscale processes (Wengel et al., 2021), which complicates the understanding

of the fate of ENSO under greenhouse gas increase, but also under AMOC changes.





The reversed pathway, i.e. ENSO impacting the AMOC, likely also exists, but also depends on several atmosphere-ocean processes which may not be adequately resolved in models. A relatively robust teleconnection exists between an El Niño event and the negative phase of the North Atlantic Oscillation (NAO) in (late) winter (Ayarzagüena et al., 2018; Brönnimann et al., 2007). The statistical relationship between the AMOC and the NAO in CMIP models depends on the subpolar North Atlantic

background state; the AMOC is less sensitive in models that have extensive sea ice cover in deep-water formation areas, while in models with less sea-ice cover, the background upper ocean stratification largely determines how sensitive the AMOC reacts to surface buoyancy forcing (Kim et al., 2023). As for ENSO, the unbiased representation of the North Atlantic mean state represents a significant challenge for CMIP models, in part due to insufficient resolution of meso-scale ocean eddies.

### 2.5.2  Influences of ENSO on the Amazon rainforest

The frequency and amplitude of ENSO variability have changed on decadal to centennial timescales in the past (Cobb et al., 2013). In recent years, extreme El Niño events combined with global warming have become increasingly associated with unprecedented extreme drought and heat stress across the Amazon basin (Jiménez-Muñoz et al., 2016), leading to increases in tree mortality, fire and dieback (Nobre et al., 2016). Imposing the surface temperature pattern of a typical El Niño event in a global atmosphere-vegetation model suggests increased drought and warming in the Amazon rainforest region (Duque-Villegas

et al., 2019), which could enhance rainforest dieback and transition to degraded and fire-prone, savanna-like ecosystems in some regions.

### 2.5.3  Influences of ENSO on the WAIS

Recent significant surface melt events on West Antarctica were associated with strong El Niño phases (Scott et al., 2019; Nicolas et al., 2017). It has been proposed that these melt events were caused by atmospheric blocking, eventually leading

to warm air temperature anomalies over West Antarctica that pass the melt point of parts of the ice sheet (Scott et al., 2019). Using reanalysis data, satellite observations and hindcasting methods, strong indications have been found that the Ross and Amundsen Sea Embayment regions are most affected by El Niño phases (Scott et al., 2019; Deb et al., 2018). In addition, it has been observed that, while ice shelves experience an increase in height (because accumulation height gains exceed basal melt height losses), they suffer from a decrease in mass (because basal ice loss exceeds ice gain from accumulation) due to increased

ocean melting during significant El Niño occurrences in the Amundsen and Ross Sea area (Paolo et al., 2018). Further, it is important to note that El Niño phases are not immediately transferred to surface melting in Antarctica but only after some time lag on the order of months (Donat-Magnin et al., 2020).

Taken together, this adds to growing body of literature that a disintegration of the West Antarctic Ice Sheet, especially along the Ross-Amundsen sector, would be favored by strong El Niño phases and tipping risks may increase if El Niño phases would

become more frequent or intense under ongoing climate change (Cai et al., 2021; Wang et al., 2017; Cai et al., 2014). This may be concerning in particular because the Amundsen region is where the most vulnerable glaciers of the West Antarctic Ice Sheet are located such as the Pine Island or Thwaites glacier (Favier et al., 2014; Joughin et al., 2014).





### 2.5.4 Influences of ENSO on warm-water coral reefs

ENSO drives abnormally high sea temperatures (seasonal heat waves above summer maxima baselines), which are super-
imposed on already warming oceans. Anomalous heat destabilizes the relationship between host corals and their symbiotic
dinoflagellate algae (zooxanthellae), resulting in severe bleaching and mortality across multiple species of corals on spatial
scales exceeding thousands of kilometers. While ENSO is geographically modulated by other ocean dipoles (e.g. Atlantic
oscillation, Indian Ocean) (Houk et al., 2020; Krawczyk et al., 2020; Zhang et al., 2017), the Pacific signal is dominant and
El Niño warm phases have been related to global episodes of extreme heat stress since the 1970s, e.g. 1979/1980, 1997/98
and 2014-2017 (Krawczyk et al., 2020; Muñiz-Castillo et al., 2019; Lough et al., 2018; Le Nohaïc et al., 2017). As global
warming progresses and oceans become significantly warmer, the incidence of mass bleaching is decoupling from El Niño
warm phase (Veron et al., 2009), with warmer conditions compared to three decades ago (McGowan and Theobald, 2023;
Muñiz-Castillo et al., 2019). The global recurrence of bleaching has reduced to an average of 6 years (Hughes et al., 2018),
sooner than expected from climate models and satellite-based sea temperatures. With warming temperatures and shortened
intervals between major bleaching, multiple human stressors, ocean acidification, and decreasing resilience, the recovery time
for mature assemblages of corals is now insufficient across most regions (Hughes et al., 2018). At the scale of the Great Barrier
Reef the emission of volatile sulfur compounds by corals adds to the local atmospheric aerosol load, increasing low level cloud
albedo and reducing warming (Jackson et al., 2018). This breaks down during physiological stress and bleaching, potentially
reinforcing thermal stress in a positive feedback loop. The potential contribution of this biologically-derived feedback loop
on local clouds, sea surface temperature and coral bleaching is uncertain however, needing validation in other locations, and
determination of any contribution to climatic conditions at larger spatial and temporal scales. While recovery from repeated
bleaching events has been observed (Palacio-Castro et al., 2023; Obura et al., 2018), the thresholds of global mean warming of
1.5°C (70-90% loss of coral reefs globally) and 2°C (90-99% loss) appear to still hold (Lough et al., 2018; Schleussner et al.,
2016; Frieler et al., 2013).

### 2.6 Effects of AMOC and ENSO changes on tropical monsoon systems

Future climate projections show a weakening of the AMOC, which can be substantial in its impact on the regional and global
climate via ocean-atmosphere connection (Pörtner et al., 2019). Evidence from modeling and paleo-reconstructions have shown
interhemispheric, low-high latitude, climate connections via ocean-atmosphere linkage for heat and moisture transport (e.g.
Nilsson-Kerr et al., 2022; Orihuela-Pinto et al., 2022a; Clemens et al., 2021; Shin and Kang, 2021).
Indeed, model simulations in response to a hosing in the North Atlantic show a clear southward shift of the ITCZ in response
to the AMOC weakening and decrease in northward oceanic heat transport (Defrance et al., 2017; Swingedouw et al., 2013;
Stouffer et al., 2006). This shift of the ITCZ impacts the various monsoon systems worldwide (Chemison et al., 2022), as
also visible in paleorecords (e.g. Sun et al., 2012). For example, Nilsson-Kerr et al. (2019) compiled paleo-reconstructions of
Heinrich stadial (11) of the penultimate deglaciation between 135 and 130 thousand years ago, suggesting an increase in the
transport of latent heat from the southern hemisphere (SH) to the northern hemisphere (NH), causing transient warming in the





NH (termination II interstadial, TII IS) and an increase in Indian Summer Monsoon rainfall. This transient warming facilitated the NH ice sheet melting which then might have reduced or shut down the AMOC, causing cooling of the NH and East Asia and a subsequent reduction of the length of the monsoon rain season (e.g. Wassenburg et al., 2021). Mechanistically, reduction of the AMOC strength either via warming and induced ice sheet melting or increased Eurasian/Arctic river runoff (e.g. Zhang

et al., 2013) cools the NH and shifts the ITCZ southward (Chemke et al., 2022), affecting spatial rainfall patterns, distribution, and amount of rainfall in the NH semi-arid and tropical monsoon regions of India and Asia.

An AMOC weakening has also been shown to strengthen the Indo-Pacific Walker circulation via cooling of the equatorial Pacific and warming of the SH/Antarctic climate on a multidecadal timescale (e.g. Orihuela-Pinto et al., 2022a). The observed AMOC weakening during the last multiple decades might be partially affected by interannual ocean-atmosphere interactions,

such as the ENSO. These superimposed effects, operating across timescales, alter relationships between the ENSO and tropical monsoon, thereby, regional rainfall patterns in a warmer climate (Mahendra et al., 2021; Pandey et al., 2020). For example, while the linear relationship between ENSO and the Indian Summer Monsoon rainfall has weakened, the ENSO-West African Monsoon relationship has increased in recent decades (Srivastava et al., 2019).

These relationships need to be further tested in paleoclimate reconstructions from both warm and cold climate states to gain

a better understanding of how an abrupt change in AMOC may have an effect on ENSO and/or on tropical monsoon systems. This would allow for a more robust predictability of tropical monsoon rainfall patterns in the future. Overall the pattern of monsoon system changes in response to tipped elements depends on the respective monsoon system.

## 2.7   Effects of permafrost regions on the global hydrological cycle

The permafrost regions have accumulated substantial amounts of ice in the soils. With ground ice melting away in a warmer

climate, permafrost landscapes experience drastic hydrological changes. The presence of ice modulates the thermophysical soil properties as well as infiltration rates and the vertical and lateral movement of water through the ground, which is often poorly represented in current Earth system models and therefore exhibits large inter-model differences. Hence, uncertainty exists about high-latitude regions becoming wetter or drier in the future. They could turn into a wetter and cooler state with many freshwater systems and lakes, which support increasing land-atmosphere moisture recycling and cloud cover, reducing ground

temperatures; or a drier state as newly formed lakes could drain, less moisture recycling supports less cloud cover and a warmer surface (Nitzbon et al., 2020; Liljedahl et al., 2016). Which parts of the Arctic will be wetter or drier in the future is uncertain, but the differences between the potential Arctic hydroclimatic futures could be very pronounced. As recently shown by de Vrese et al. (2023), the drier and warmer permafrost state would lead to less sea ice, a reduced pole-to-equator temperature gradient, and a weaker AMOC. The drier state has more boreal forest extended to the north, while higher frequency and extent of forest

wildfires. In comparison with the wetter state, the drier Arctic state also shifts the position of the Intertropical Convergence Zone which results in higher precipitation in the Sahel region and potentially also in the Amazon rainforest region. Increased forest and vegetation cover in these regions would be the consequence (de Vrese et al., 2023). Therefore, shifts in permafrost hydrology could affect climate tipping elements far beyond Arctic boundaries. Insofar the hydrological cycle due to Permafrost changes may have far-reaching impacts.





## 2.8 Interactions between multiple tipping elements and planetary cascades

Assembling the individual links mentioned before in sections 2.2–2.7 gives rise to the possibility of tipping cascades involving more than two elements. These could lead to large changes at the regional and even planetary scale. A plausible example are Dangaard-Oeschger (D-O) events (section 3.2.2). Another example comes from the study of the last interglacial period, for which proxies for sea ice, polar ice sheets, AMOC, boreal forest, and permafrost indicate abrupt changes (Thomas et al., 2020). Although the dating uncertainties make it difficult to determine the causal structure of a potential cascade, positive feedbacks between these TEs could explain the amplified polar temperatures and sea level rise obtained from reconstructions (+8°C in Greenland, and +6-9m sea level rise compared to today) (Dutton et al., 2015; NEEM community members, 2013).

On a larger scale, tipping cascades could be responsible for driving the Earth system into completely different climatic states that have been identified in paleo-data (Westerhold et al., 2020), climate models of intermediate complexity (Lucarini and Bódai, 2017) and general circulation models at coarse spatial resolution (Brunetti et al., 2019; Popp et al., 2016; Ferreira et al., 2011; Voigt and Marotzke, 2010). For example, a tipping cascade involving ocean circulation and ice sheets might have been responsible for a transition from a greenhouse to an icehouse state at the Eocene-Oligocene boundary (section 3.1) . A major concern regarding the future may be that a cascade involving several tipping elements and feedbacks could lock the Earth system in a pathway towards a hothouse state with conditions resembling that of the mid-Miocene (+4-5°C, +10-60m sea level compared to the pre-industrial baseline) (Burke et al., 2018; Steffen et al., 2018). Feedbacks that affect global temperature could involve albedo changes (through ice sheet or sea ice loss) and additional $CO_2$ and $CH_4$ emissions (through permafrost thawing, methane hydrates release). In the worst case (and unlikely) scenario, a single tipping event could propel the Earth system into such a hothouse, for example the hypothesized breakup of stratocumulus decks (Schneider et al., 2019).

Time scales are crucial when discussing hothouse scenarios. A potential hothouse state in the next centuries seems implausible in light of the current state of research. For example, in climate projections up to 2100, CMIP6 models show no evidence of non-linear responses on the global scale. Instead, they show a near-linear dependence of global mean temperature on cumulative $CO_2$ emissions (Masson-Delmotte et al., 2021). Similarly, in a recent assessment, Wang et al. (2023) concluded that a tipping point cascade with large temperature feedbacks over the next couple of centuries remains unlikely and that while the combined effect of tipping elements on temperature is significant for those time scales, it is secondary to the choice of anthropogenic emissions trajectory. However, this does not completely rule out the possibility of a hothouse scenario in the longer term. Indeed, tipping events are not necessarily abrupt on human time scales. Positive feedbacks could have negligible impacts by 2100, for example on global mean temperature and sea level rise, but still influence Earth system trajectories on (multi-) millennial time scales (Kemp et al., 2022; Lenton et al., 2019; Steffen et al., 2018). Overall, this calls for experiments across the model hierarchy. EMICs (Earth System Models of Intermediate Complexity) in particular, and AOGCM at coarse spatial resolution, offer an interesting trade-off as they include representations of most tipping elements while still allowing for multi-millennial simulations.



## 3 Archetypal examples of interactions between tipping elements from a paleoclimatic perspective

### 3.1 Interactions in the distant past: Eocene-Oligocene Transition

The formation of a continent-scale ice sheet on Antarctica during the Eocene-Oligocene Transition about 34 million years ago
is known as Earth's Greenhouse-Icehouse Transition. Following a cooling over tens of millions of years, this shift to a new
climate state would have been visible from space, as Antarctic forests were replaced by a blanket of ice, and seawater receded
from the continents, changing the shapes of coastlines worldwide. The climate transition is recorded as a shift in the oxygen
isotopic composition of microscopic fossil shells in marine sediment cores, which reflects a combination of deep sea cooling
and continental ice growth (Coxall et al., 2005). It had global consequences for Earth's flora and fauna, both in the oceans and
on land (Hutchinson et al., 2020).

This climate transition has been identified as a possible palaeoclimate example of cascading tipping points in the Earth
system (Dekker et al., 2018; Tigchelaar et al., 2011). Examples of climatic tipping elements in this case consist of global deep
water formation, the Antarctic Ice Sheet, polar sea ice, monsoon systems and tropical forests. In a conceptual model, the first
part of the oxygen isotope shift is attributed to a major transition in global ocean circulation, while the second phase reflects
the subsequent blanketing of Antarctica with a thick ice sheet (Tigchelaar et al., 2011).

The global ocean circulatory system was showing tentative signs of change a few million years before the climate transition,
likely caused by changing ocean gateways in the north Atlantic (Coxall et al., 2018). Neodymium isotopes do suggest that a
precursor to North Atlantic Deep Water reached the southern hemisphere close to the Eocene-Oligocene Transition, perhaps
signaling the onset of Atlantic Meridional Overturning Circulation (AMOC) (Via and Thomas, 2006). However, the exact
timing remains uncertain and may not correlate to the onset of the oxygen isotope shift. Indeed, the first part of the isotope
shift is associated with a cooling of both deep sea temperatures and low latitude sea surface temperatures, which therefore
more likely reflects a change in radiative forcing (Kennedy et al., 2015; Lear et al., 2008). However, this does not preclude
AMOC onset preconditioning the system for glaciation through heat piracy in the Southern Ocean, with the exact timing of the
transition set later by a favorable orbital configuration (Coxall et al., 2005).

In general, biomes in Earth's greenhouse state reflect warmer and wetter conditions than the icehouse state of the early
Oligocene, but many of these seemed to have changed gradually as climate cooled in the Eocene, making it difficult to identify
vegetation tipping elements following the glaciation of Antarctica (Hutchinson et al., 2020). The mammalian fossil record,
which is coupled to vegetation through diet, suggests more acute changes in the early Oligocene. The Grand Coupure (="The
Big Break"), is a long-known mammalian extinction/origination event during the Eocene-Oligocene time involving large scale
migrations of Asian mammals into Europe (Hooker et al., 2004). Thought to signal a combination of changing climate and flo-
ral changes, this abrupt faunal turnover might reflect crossing of an ecosystem tipping point caused by the crossing of a climate
tipping point: a climate-ecology tipping cascade. Mammal extinctions seem to be particularly widespread in Afro-Arabia and
linked to loss of dietary diversity (de Vries et al., 2021). This finding is consistent with the idea that biomes in this subtrop-
ical region are tippable elements (Armstrong McKay et al., 2022; Lenton et al., 2008). Other evidence of vegetation biomes
having tipped includes a transition from warm-temperate to cool-temperate rainforests in southeastern Australia (Korasidis




et al., 2019). Monsoon systems, sensitive to forcing and to large-scale reorganizations of the climate system, might have been important for explaining the shifts in these respective vegetation biomes. Moreover, simulations of the late Eocene climate suggest the existence of a strong monsoon-like climate over the Antarctic continent; without a major reorganization of such an atmospheric circulation regime, ice growth on Antarctica seems very unlikely (Baatsen et al., 24 May 2023, 2020).

The glaciation of Antarctica also produced a sea level fall of several tens of meters (Lear et al., 2008), causing shallow seaways to recede, turning many marine regions into continental habitats, which experienced particularly strong seasonality (Toumoulin et al., 2022). The associated reduction of the marine carbonate factory in previously submerged tropical shelf-seas, caused the calcite compensation depth to deepen by more than one kilometer, turning enormous swathes of seafloor white as the sinking calcite shells of plankton no longer dissolved in shallow depths (Coxall et al., 2005).

In summary, Earth's Greenhouse-Icehouse Transition was likely associated with a range of interactions between components of the Earth system that are debated as potential tipping elements. Determining the extent to which these reflect a cascading series will require a major data-modeling effort, with improved correlations between marine and terrestrial records, and better constraints on the rate and magnitude of change within a range of tipping elements.

### 3.2 Interactions during and since the last glacial period

In this chapter, we discuss three important paleoclimate candidates for tipping interactions since the last glacial period (see Fig. 3).

#### 3.2.1 Bølling-Allerød

Towards the end of the last ice age, a very prominent event is recorded in numerous geological archives. The Bølling-Allerød (B/A) started at 14.7 ka with abrupt warming in the Northern Hemisphere (with temperature increase in Greenland by 10-14°C
over a few years; Andersen et al. (2004)) in response to a reinvigoration of the AMOC (McManus et al., 2004) and lasted until 12.9 ka. The B/A is an example of pronounced interactions between Earth system components and cascading impacts in the Earth system (Brovkin et al., 2021). At the onset of the B/A, atmospheric $CO_2$ and $CH_4$ concentrations rapidly increased over a few decades (Marcott et al., 2014) in response to abrupt warming and permafrost thaw (Köhler et al., 2014) and moisture changes (e.g. Kleinen et al., 2023). This was followed by fast changes in vegetation composition (Novello et al., 2017; Fletcher
et al., 2010). In the ocean, surface warming and circulation changes were propagated downward, leading to sedimentary anoxia across the North Pacific (Praetorius et al., 2015; Jaccard and Galbraith, 2012) as well as more severe hypoxia in the Cariaco Basin (Gibson and Peterson, 2014) and Arabian Sea (Reichart et al., 1998), indicating a link between climate warming and ocean deoxygenation.

The trigger for the rapid amplification of ocean circulation and the associated abrupt impacts at the B/A transition has been
a focus of debate, with opinions divided between an essentially linear response to the (possibly abrupt) cessation of freshwater forcing (e.g. Liu et al., 2009) versus a non-linear response to more gradual forcing (i.e., a tipping point, e.g., Barker and Knorr (2021); Knorr and Lohmann (2007)).





Gradual changes observed in key climatic variables (e.g., $CO_2$ and global temperature) during sustained periods of cold across the surface North Atlantic (as occurred prior to the B/A onset) were a persistent feature of glacial terminations through-
out the last 800 kyr (e.g. Barker et al., 2019), as well as during the massive ice rafting events of the last glacial period (known as Heinrich events). Each of these periods is thought to have been followed by the rapid resumption of ocean circulation and other events associated with the B/A (e.g., a rapid rise of $CO_2$ and $CH_4$).

### 3.2.2   Dansgaard-Oeschger events

Smaller amplitude (as compared to B/A) yet equally rapid, transitions known as Dansgaard-Oeschger (D/O) events (Fig. 3)
occurred repeatedly during glacial periods throughout much of the late Pleistocene (e.g. Ganopolski and Rahmstorf, 2001). In general, these consist of an abrupt (on the order of decades) warming from stadial to interstadial conditions, followed by gradual cooling over the course of hundreds to a few thousands of years, before a rapid transition back to cold stadial conditions. Evidence from Greenland ice cores and North Atlantic sediments suggest that the abrupt cooling transitions (from warm interstadial to cold stadial conditions) were systematically preceded and possibly triggered by more gradual cooling
across the high latitude Northern Hemisphere (e.g., NGRIP project partners; Barker et al. (2015)). The abrupt transitions from stadial to interstadial conditions were also preceded by more gradual changes elsewhere (for example increasing Antarctic and deep ocean temperatures and decreasing dustiness; Barker and Knorr (2007)), leading to the idea that both types of transitions may be predictable to some extent (Lohmann, 2019; Barker and Knorr, 2016). Each event was also paired with rapid changes in ocean circulation, terrestrial hydroclimate, atmospheric composition and ocean oxygenation in much the same way as observed
during the B/A. Thus, the occurrence and interactions among many subsystems that show abrupt changes make it plausible to consider it a cascade, and are a ubiquitous and common feature of late Pleistocene climate variability.

During the abrupt warming phases of D/O cycles, an abrupt decrease of Arctic and North Atlantic sea ice cover likely contributed to the onset of convection and a rapid resurgence of a much weaker, and potentially even collapsed, AMOC (Gildor and Tziperman, 2003; Li et al., 2010). D/O-type changes in coupled climate models also feature a rapid disappearance
of sea ice that precedes the abrupt AMOC strengthening (Vettoretti and Peltier, 2016; Zhang et al., 2014). Thus, the D/O warmings may potentially comprise a tipping cascade (Lohmann and Ditlevsen, 2021). However, such a cascading interaction may depend on the climate background state, and it is unclear whether North Atlantic sea ice cover during the last glacial period can be considered a tipping element.

### 3.2.3   Heinrich events

While the exact causes and mechanisms of the B/A transition and D/O events are still under debate, Heinrich events are better understood. They occurred during some of the cold stadial phases mentioned above and were associated with major reorganization of ocean circulation in the North Atlantic (for a review see Clement and Peterson (2008)). During Heinrich events, large masses of ice were released from the Laurentide Ice Sheet, leading to a dramatic freshening of the North Atlantic Ocean and enhanced suppression of deep-water formation and the AMOC.





**Figure 3.** Interactions at the end of Heinrich Stadial 4 (HS4). (a) Climate proxy indices spanning the transition from HS4 into Dansgaard-Oeschger (DO) event 8 (time goes from left to right). From top to bottom: AMOC strength (Henry et al., 2016), Norwegian Sea ice cover (Sadatzki et al., 2020), Greenland temperature (North Greenland Ice Core Project members (NGRIP), 2004), North Atlantic SST (Martrat et al., 2007), Dust accumulation in Greenland (Ruth et al., 2007), Asian monsoon intensity (Cheng et al., 2016), South American monsoon intensity (Kanner et al., 2012). Horizontal red bar indicates period when ITCZ assumed a more southerly position (Wang et al., 2004). Hatched region spans the transition from HS4 to DO8 and represents an estimate of the relative age uncertainty among the records shown (i.e. it is generally not possible to tell which changes occurred earlier or later within the overall sequence). Vertical arrows indicate the sense of increase for each parameter. (b) Interactions between Ocean, Atmosphere and Land during the end of HS4.



They can be understood as a phenomenon involving two tipping elements, the Laurentide Ice Sheet and the AMOC. The ice fluxes from the Laurentide Ice Sheet have been described as a binge/purge oscillator (MacAyeal, 1993), where a period of strong ice accumulation (the binge phase) is followed by a period of rapid ice loss (the purge phase). During the binge phase, ice is generally thought to be frozen to the bottom and thus immovable. As the ice sheet gets thicker, basal temperatures increase until the pressure melting point of the basal ice is reached. The resulting meltwater production lubricates the bed,

and enables sliding of the ice. This may already be sufficient to initiate the purge phase, though further triggers like ocean subsurface warming probably also played an important role in destabilizing marine-terminating portions of the Laurentide Ice Sheet (Max et al., 2022; Alvarez-Solas et al., 2013). The purge phase lasts until the ice sheet has become too thin to maintain basal temperatures above the pressure melting point, thus re-freezing and stopping the ice flow. The resulting ice stream flows into the Atlantic Ocean, and as the resulting icebergs melt, Atlantic surface waters are freshened to the point where the AMOC

cannot be sustained and collapses.

The mechanisms sketched above have been demonstrated in a number of transient model experiments, using Earth System Models of Intermediate Complexity (Calov et al., 2010, 2002) and complex ice-sheet-atmosphere-ocean general circulation models (Schannwell et al., 2023; Ziemen et al., 2019). However, not all details have yet been resolved, the exact trigger mechanism (and threshold) initiating the purge phase, for example, has not yet been identified (Schannwell et al., 2023).

### 3.3   A paleoclimate perspective on the resilience of the Amazon rainforest

Two historical analogues have provided some (albeit not fully consistent) insights into the response of the Amazon to reductions in rainfall: Heinrich events during the last glacial period, and the mid-Holocene. As mentioned in the previous section, Heinrich events are remarkable intervals during the last glacial period in which the AMOC seems to have substantially weakened in response to iceberg release in the North Atlantic (Henry et al., 2016). Paleoclimate data from these events are of great help to

evaluate the processes suggested by climate model simulations of AMOC slowdown. Häggi et al. (2017), using an isotope proxy from a sediment core collected offshore the Amazon River mouth, showed savannah intrusions into the Amazon rainforest during repeated Heinrich events. The intrusions of savannah occurred in northern Amazonia (Zular et al., 2019; Häggi et al., 2017) and validate the suggested decrease in precipitation over that region in response to AMOC weakening (Campos et al., 2019). Further precipitation and, even more importantly, vegetation reconstructions with appropriate age models and sufficient

temporal resolution, will help clarify the southward extent of the drying of northern Amazonia due to an AMOC collapse, as well as its consequences to the rainforest.

Kukla et al. (2021) used pollen, charcoal, and speleothem oxygen isotope proxy data to reconstruct the response of the Amazon forest during the mid-Holocene, when precipitation was relatively low (Prado et al., 2013). Their analysis suggests that the Amazon was resilient to rainfall reductions as high as projected by climate models for the rest of the century. However,

it also has to be considered that in the study of Kukla et al. (2021) temperature and land use were similar to pre-industrial conditions, whereas future warming and deforestation will act as additional stressors that affect the surface water balance by increasing potential evapotranspiration and decreasing precipitation recycling, on one hand (Zemp et al., 2017), while also increasing the chances of fire and thus the possibility of the Amazon to convert into a degraded, open ecosystem





## 4 Archetypal example of nonlinear climate component interactions: Arctic sea ice loss leading to coastal permafrost
### erosion

As even regional tipping elements can have substantial spatial extent it is possible that only a part of a tipping element or nonlinearly behaving region is affected. Here we discuss an example of such nonlinear climate component interaction, namely the impact of accelerating Arctic sea ice loss on coastal erosion in Siberian and North American permafrost regions.

Relic carbon-rich coastal shelf ice in Siberia and Alaska is exposed to the ocean and atmosphere (Irrgang et al., 2022). Currently, Arctic coastal erosion rates are of 4 m a$^{-1}$, peaking at 25-50 m a$^{-1}$, an order of magnitude higher than elsewhere in the world (Philipp et al., 2022). The shortened sea ice season exposes the ocean to winds and increases ocean wave fetch, leading to higher ocean waves that accelerate erosion levels (Meucci et al., 2023; Hošeková et al., 2021). The Siberian and Alaskan coasts transition to a higher erosion state, with a total retreat of 500-1000 m since the 1950s, and an accelerated retreat since the mid 2000's (Figure 4) (Grigoriev, 2019).

The shelf primary production sustains about a third of Arctic production; coastal erosion provides the majority of nitrogen and phosphorus and a third of carbon (Terhaar et al., 2021). These fluxes can be much higher in the future, changing marine food-webs. Summer sea ice disappearance increases seasonal bloom and nutrient depletion in the ocean, nutrient inputs from rivers and coastal erosion alleviate the nutrient limitation (Oziel et al., 2022). The erosion is a risk to infrastructure, settlements and economy (Clare et al., 2022). The impact of erosion on ecosystems is medium to high with the medium to high uncertainty.

Change in sea ice is gradual (Notz, D. and SIMIP Community, 2020), however, storms can abruptly change sea ice on the shelves (Lukovich et al., 2021), leading to high waves and a destabilization of parts of the permafrost coast (Casas-Prat and Wang, 2020). The impact of coastal erosion on ecosystems is irreversible, as are socioeconomic impacts (Fritz et al., 2017). Furthermore, coastline collapse and permafrost degradation can release large amounts of carbon to the ocean and atmosphere (Vonk et al., 2012; Tarnocai et al., 2009).

In summary, this cascade operates as follows: (1) abrupt changes (a tipping point) in summer-autumn sea ice retreat from the coast leads to (2) increase in the waves, resulting in (3) abrupt increases in erosion rates (2-4 times higher) due to a wave undercut mechanism. Thus, (4) there is a potential cascading risk of large carbon releases to the ocean and atmosphere due to the coastal collapse. At the same time, coastal ecosystems would be impacted through increased nutrients and other terrigenous matter fluxes as well as local communities and economies (fisheries and infrastructure collapse).

## 5 Modeling tipping element interactions and cascading transitions

Modeling interactions between tipping elements and potential tipping cascades in the climate system is a difficult task. A key challenge is to accurately capture feedback mechanisms between different climatic components. In addition, each climate sub-system evolves over spatial scales and time scales that can span orders of magnitude from decades to centuries for biosphere components, and from centuries to millennia for the large ice sheets on Greenland and Antarctica (see Fig. 2). The ideal tool to study the interaction between tipping elements would therefore be a high-resolution, comprehensive Earth system model based on general circulation models (GCM) for the atmosphere and ocean, with a sea ice component, dynamic vegetation and



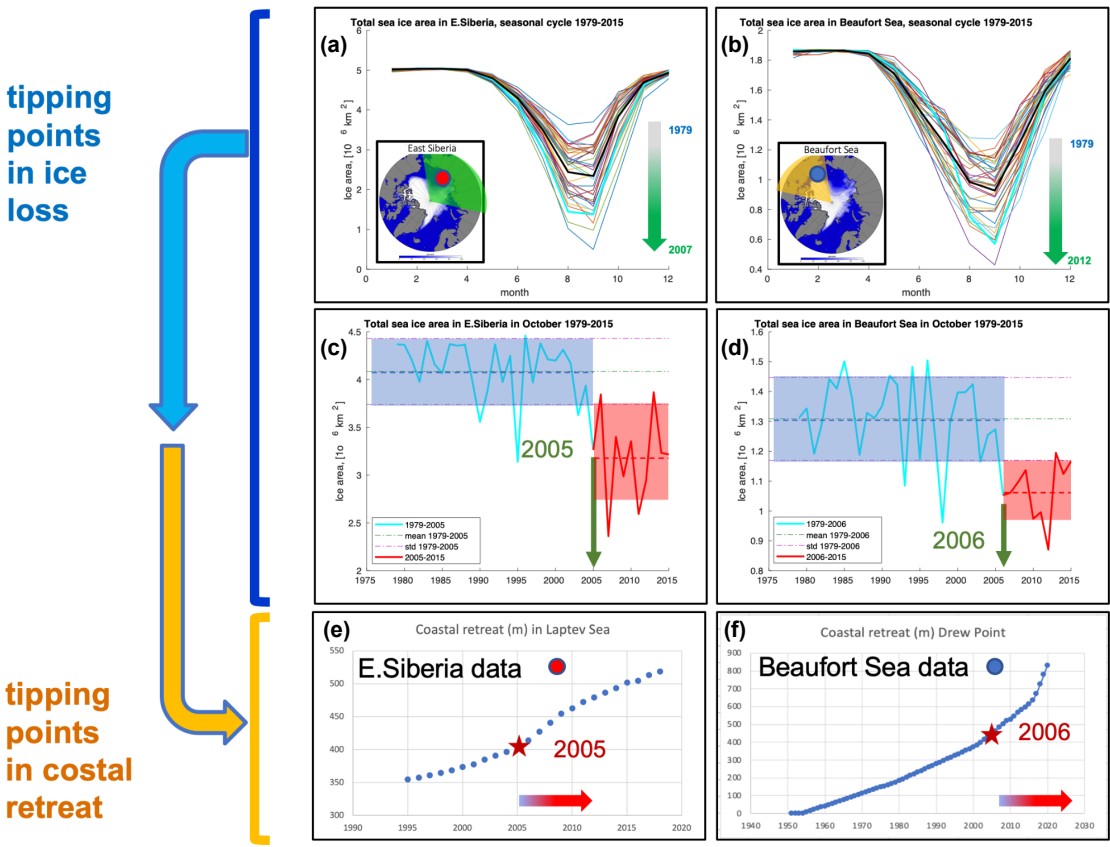

**Figure 4.** Cascade between different elements of the Arctic climate system: sea ice and coastal permafrost erosion. (a,b) Sea ice area seasonal cycle for the 1979-2015 in Siberia and Beaufort Sea. (c,d) Time series of the October sea ice area with a tipping point, defined through changes in the mean, standard deviation and linear trends; the two ice states are marked by the cyan and red lines. (e,f) Coastal retreat in these regions. Stars show potential tipping points. The mechanism of transition, linking the sea ice retreat to the increased waves and accelerated coastal erosion, suggests a cascade, acting from the abrupt changes in the ocean and cryosphere to the changes of state in the coastal retreat and ecosystem.





interactive ice sheets and carbon cycle. Moreover, the model should be computationally fast to allow the representation of the slow processes and to run comprehensive ensembles for taking into account the uncertainties in key parameters (Murphy et al., 2004). However, such a universal tool does not exist. Instead, a hierarchy of models of different complexity is needed to explore

the interactions between tipping elements on different temporal and spatial scales (Fyke et al., 2018). The state-of-the-art Earth system models that are usually employed in climate change projections, the models in the Coupled Model Intercomparison Project 6 (CMIP6) (Eyring et al., 2016), are the first choice to investigate processes developing over centennial time scales. Since these models usually include dynamic vegetation, they are suitable to explore the interactions between abrupt changes in ocean circulation, Arctic sea ice and vegetation cover. However, CMIP6 models are computationally expensive and most of

them do not include interactive ice sheets, which limits the applicability of these models to study interactions between tipping elements and potential tipping cascades that involve slow deep-ocean dynamics or ice sheets. Also, they show some limitations to how vegetation is represented, especially in tropical areas (e.g. D'Onofrio et al., 2020). Recently, progress has been made in including interactive ice sheets in a few CMIP6 models for studying the coupled climate-Greenland evolution (Madsen et al., 2022; Ackermann et al., 2020; Muntjewerf et al., 2020) on centennial time scales.

For studying feedbacks on millennial time scales or longer, one possible solution is to use Earth system models with coarser spatial resolution, allowing for faster simulations (Brunetti and Ragon, 2023; Brunetti et al., 2019; Ferreira et al., 2011; Hawkins et al., 2011) or Earth System Models of Intermediate Complexity (EMICs, Claussen et al. (2002)). A downside of these models is that the interactions between tipping elements are necessarily less realistic and some nonlinear processes need to be parameterized at sub-grid level. In particular, EMICs are faster than GCMs of comparable spatial resolution, since

they make use of some approximations in the representation of the atmosphere and/or ocean dynamics, and can for instance be applied to investigate climate-ice sheet interactions on multi-millennia time scales (Willeit et al., 2022; Quiquet et al., 2021; Choudhury et al., 2020).

    An alternative technique to speed up complex models, and therefore enable them to explore feedbacks on longer time scales, is offline (asynchronous) coupling, which has been applied to represent vegetation-climate (Betts et al., 1997; De Noblet

et al., 1996; Claussen, 1994) and ice sheets-climate interactions (Scherrenberg et al., 2023; Pohl et al., 2016; Herrington and Poulsen, 2011; Pollard, 2010). One example of asynchronous coupling is when the atmosphere evolves with fixed vegetation cover and eventually the latter is updated to the equilibrium conditions of the former (Foley et al., 1998). However, despite being less computationally expensive, feedback mechanisms and thus tipping phenomena and cascades are better represented when dynamical (synchronous) coupling is implemented (Drüke et al., 2021; Fisher et al., 2018; Fyke et al., 2018; Bonan et al.,

585   2003).

    Tipping phenomena and cascades at the regional scale, however, may be investigated with different approaches (Bastiaansen et al., 2022). For instance, they can be investigated using observation-based simulations coupled to energy or hydrological balance models at high spatial resolution as has been done for tipping cascades in the Amazon rainforest (Wunderling et al., 2022). Regional implementations of ice shelf/ocean interaction exist to obtain improved estimates of basal melt and to include

small scale processes, like the presence of ocean eddies (Dinniman et al., 2016) that can affect the overall stability of the system and potentially intensify transitions or cascades. It is indeed possible to run regional climate models (RCMs) (Noël et al., 2018;





Rae et al., 2012) at horizontal grid resolutions of a few km, thus providing more accurate spatio-temporal distributions of climatic variables like precipitation and temperature than GCMs. An alternative modeling framework is to apply grid refinement over a specified region of interest in a GCM, which avoids inconsistencies between the different dynamical cores and physical

parameterizations used in RCMs and GCMs, and (more importantly for tipping phenomena or casades) allows for two-way interactions between the refinement region and the global domain (Van Kampenhout et al., 2019). With such regionalized modeling approaches it could be possible to empirically detect local to super-regional (cascading) regime shifts (Rocha et al., 2018).

Also, more conceptual approaches based on differential equations or box models are frequently used for studying tipping
events and cascades for present-day climate and paleoclimates (Lohmann et al., 2021; Wunderling et al., 2021b; Wood et al., 2019; Boers et al., 2018). While such models offer a unique way of unraveling the complex dynamics of interacting tipping elements, it is not guaranteed that results of conceptual models can be confirmed by complex models, since the formers consider only a limited subset of dynamical variables and nonlinear processes of the climate system (Bathiany et al., 2016). For example, simple models that do not include space are suggested to over-predict the occurrence of tipping points, while
spatial pattern-formation phenomena might prevent such tipping when space is explicitly included (Rietkerk et al., 2021).

Lastly, modeling of cascading effects from the physical system to society and economy as well as vice versa are still missing from most state-of-the-art Earth system models, requiring urgent development (Franzke et al., 2022; Steffen, 2021; Beckage et al., 2020)

## 6 Discussion & Conclusion

As anthropogenic global warming continues, tipping elements are at risk of crossing critical thresholds (Armstrong McKay et al., 2022). Several assessments have investigated the risk of crossing critical thresholds of individual tipping elements whereas interactions between tipping elements are only more recently taken into account, mostly by conceptual models (e.g. Sinet et al., 2023; Wunderling et al., 2023; Dekker et al., 2018). In this review, we summarize the current state of the literature of many central tipping element interactions. We conclude that tipping elements interact across scales in space and time (see
Figs. 1 and 2), spanning from subcontinental to nearly planetary spatial scales from sub-yearly up to millennial time scales. We find that many of the discussed interactions between tipping elements are of destabilizing nature (Tab. 1), implying the possibility of cascading transitions under global warming. Out of the discussed interactions, nine are assessed as destabilizing while two are stabilizing, and three of unclear status/sign (see Fig. 1, without dashed arrows). Assessing the overall stability of the Earth system, and the possibility of a chain of nonlinear transitions, will however require more detailed assessments of
interactions, their effect strengths, time scales and state-dependence.

While there is more and more research on individual thresholds of climate tipping elements, substantial uncertainties prevail in the existence and strength of many links between tipping elements. In order to decrease such uncertainties, we propose four possible ways forward: (i) Observation-based approaches: Satellite observations, reanalysis and paleoclimate data sets may be evaluated using correlation measures (Liu et al., 2023), nonlinear approaches such as convergent cross-mapping (Van Nes



et al., 2015) or more advanced machine-learning approaches such as causal inference (e.g. Runge et al., 2019; Kretschmer
et al., 2016; Runge et al., 2015). (ii) Earth system model-based approaches: With recent progress, Earth system models of
full or intermediate complexity could be used to evaluate interactions between climate tipping elements in process-detail and
quantify their interactions using specifically designed experiments. (iii) Risk analysis approaches: Since relevant parameter
and structural uncertainties are large within Earth system models, analyzing model ensembles with a considerable number

of ensemble members is very helpful in order to comprehensively propagate uncertainties for risk assessments (Daron and
Stainforth, 2013; Stainforth et al., 2007; Murphy et al., 2004). While this approach often requires more simplified or emulator
models designed for large-scale Monte Carlo analyses, it does not reduce model or data uncertainties per se. Therefore, it is
still possible to evaluate the risk of emerging tipping events or cascades as well as the role of interactions between tipping
elements. (iv) Expert elicitation: An expert elicitation on tipping element interactions would be of tremendous value to update

and move beyond early investigations of this kind (Kriegler et al., 2009), since all the three aforementioned approaches (i-iii)
have important limitations that would benefit from direct expert input.

Clearly all of these strategies have their strengths and limitations. Both (i) and (ii) could benefit from extending the es-
tablished notion of correlation measures in climate networks (Liu et al., 2023; Ciemer et al., 2021; Armstrong et al., 2019;
Svendsen et al., 2014; Chen et al., 2010) to causal measures such as causal inference methods, for instance informed by

Granger causality or conditional (in)dependences (Pearl, 1985; Granger, 1969). A prominent approach for causal inference
has been applied in climate science using the so-called PCMCI-algorithm (Runge et al., 2019), which is a constraint-based
method that considers (lagged) partial correlation to establish links between the considered variables. Such methods can be
used to check whether identified correlations indeed correspond to a causal relation, where it is important to take into account
all possible confounding factors. To do that one needs to start from the physical processes involved, for instance informed

by conceptual models, and test this as a hypothesis for the causal relations (Kretschmer et al., 2021; Di Capua et al., 2020).
However, to apply such methods to the tipping point context it is important to know the limitations. One of the assumptions
made is that of stationarity of the links between the variables considered, which may not be true once a tipping point is crossed.
Another difficulty may be the different timescales of the tipping elements where, e.g., ice sheets are very slow compared to the
Amazon rainforest or AMOC.

Limitations of the approaches (i)-(iv) further include: First, observation-based data from recent monitoring do not include
tipping of large-scale Earth system components due to the lack of recent occurrence of such events or even cascades. Paleo-
climate data can partially compensate for such disadvantages at the price that data is hard to retrieve and its availability and
abundance is far from perfect. Second, complex Earth system models may not include all relevant interaction processes be-
tween tipping elements or are often computationally too expensive to run large-scale ensembles that could take into account

and propagate all relevant uncertainties. And third, risk analysis approaches include accounting for theoretical knowledge prop-
erties of these types of physical dynamical systems. Therefore, different approaches should complement each other, requiring
experts to combine observations, reconstructions, and novel computational strategies individually, but potentially also through
a formalized elicitation. Thus improving model development and informing risk analyses based on ensembles of possibilistic
model simulations. Taken together, all approaches mentioned above are required to obtain more reliable estimates of existen-





tial risks such as those posed by potential tipping events or even cascades (Kemp et al., 2022; Jehn et al., 2021). They could be used to inform an emulator for tipping risks taking into account properties of individual tipping elements as well as their interactions. In addition, there also exist large uncertainties not only among the known interactions as discussed above, but also not all interactions are known or quantified (known unknowns versus unknown unknowns).

Further, in certain systems, there are forcings of non-climatic origin that could interact with climate change and lead to
tipping, and thus to interactions and possibly cascades with other elements. For instance, land use change and, specifically, deforestation are threatening the Amazon and decreasing its resilience to climate change (e.g. Staal et al., 2020), since the Amazon is transpiring large parts of its own rainfall. Recent studies also indicate that Amazon and other humid forests might also affect the atmospheric convergence of moisture (Makarieva et al., 2023), which might possibly affect other climate and ecosystem elements. Therefore, non-climate related factors might also trigger cascading tipping, which would require further
research to be investigated. Lastly, systems do not necessarily tip fully in one go, but there can also be stable intermediate states (such as through the formation of spatial patterns). This has mostly been reported in ecological systems but is not limited to them (Rietkerk et al., 2021; Bastiaansen et al., 2020).

Taken together, assessing and quantifying tipping element interactions better has great potential to advance suitable risk analysis methodologies for climate tipping events and cascades, especially because it is clear that tipping elements are not
isolated systems. Clearly, the relevance for developing such risk analysis tools to assess tipping events and cascades is evident, given the potential for existential risks and long-term irreversible changes (Kemp et al., 2022).

*Code and data availability.* There is no data or code that has been produced for this review article.

*Author contributions.* N.W. and A.vdH. designed the study. N.W. and A.S.vdH. led the writing of the manuscript with input from all authors. N.W., A.S.vdH., N.J.S, Y.A., S.B., J.T.B., V.B., T.K., R.W., V.C., A.K., H.C., and C.L. designed the figures of this manuscript. All authors
have reviewed and edited the final version of the manuscript.

*Competing interests.* Some authors are members of the editorial board of the journal Earth System Dynamics. The peer-review process was guided by an independent editor, and the authors have also no other competing interests to declare.

*Acknowledgements.* This review article has been carried out within the framework of the Global Tipping Points Report 2023. N.W. and J.F.D. acknowledge support from the European Research Council Advanced Grant project ERA (Earth Resilience in the Anthropocene,
ERC-2016-ADG-743080). J.F.D. is grateful for financial support by the project CHANGES funded by the German Federal Ministry for Education and Research (BMBF) within the framework 'PIK_Change' under grant 01LS2001A. R.W. acknowledges financial support via the Earth Commission, hosted by FutureEarth. The Earth Commission is the science component of the Global Commons Alliance, a sponsored



project of Rockefeller Philanthropy Advisors, with support from Oak Foundation, MAVA, Porticus, Gordon and Betty Moore Foundation, Herlin Foundation and the Global Environment Facility. The Earth Commission is also supported by the Global Challenges Foundation. V.C. is funded as Research Fellow by the Belgian National Fund of Scientific Research (F.S.R. – FNRS). A.K.K. and R.W. acknowledge support by the European Union's Horizon 2020 research and innovation programme under Grant Agreement No. 820575 (TiPACCs) and No. 869304 (PROTECT). T.K. acknowledges support through the project Palmod, funded by the German Federal Ministry of Education and Research (BMBF), 01LP1921A. M.Br. acknowledges the financial support from the Swiss National Science Foundation (Sinergia Project No. CR-SII5_180253). Coastal and marine sections would like to thank the generous support of the BNP PARIBAS Foundation for the CORESCAM project, part of the 2019 call on Biodiversity and Climate Change, and the USAID for support to the SWAMP project. J.T.B. gratefully acknowledges the UK Research Councils funded Models2Decisions Grant (M2DPP035: EP/P0167741/1), ReCICLE (NE/M004120/1) and STFC Spark Award (ST/V005898/1) which helped fund his involvement with this work. A.S.vdH., S.K.J.F. acknowledge funding by the Dutch Research Council (NWO) under a Vici project to A.S.vdH. (with project number VI.C.202.081 of the NWO Talent programme). A.S.vdH. has worked under the program of the Netherlands Earth System Science Centre (NESSC), financially supported by the Ministry of Education, Culture and Science (OCW). A.S.vdH., R.B., S.S. acknowledge funding from the European Union's Horizon 2020 research and innovation programme under grant agreement No 820970 (this paper is TiPES paper #232) and under the Marie Skłodowska-Curie Grant Agreement No. 956170 (CriticalEarth). CHL acknowledges NERC funding for SWEET grant NE/P019102/1. JB acknowledges funding by the European Research Council (ERC) under the Horizon Europe research and innovation programme (ERC-starting grant CATES, grant agreement No. 101043214). C.M.C. acknowledges the financial support from FAPESP (grants 2018/15123-4 and 2019/24349-9) and CNPq (grant 312458/2020-7). Y.A. and S.R. acknowledge funding from the project COMFORT (grant agreement no. 820989) under the European Union's Horizon 2020 research and innovation programme, and from the EC Horizon Europe project OptimESM "Optimal High Resolution Earth System Models for Exploring Future Climate Changes", grant 101081193 and UKRI grant 10039429. Y.A. and S.R. also acknowledge funding support from the project EPOC, EU grant 101059547 and UKRI grant 10038003 and from the UK NERC projects LTS-M BIOPOLE (NE/W004933/1), CANARI (NE/W004984/1), and Consequences of Arctic Warming for European Climate and Extreme Weather (Arcti-CONNECT, NE/V004875/1). Y.A. and S.R. acknowledge the use of the ARCHER UK National Supercomputing and JASMIN. For the EU projects the work reflects only the authors' view; the European Commission and their executive agency are not responsible for any use that may be made of the information the work contains. D.D. is funded by the Belgian Science Policy Office (BELSPO) under the RESIST project (contract no. RT/23/RESIST). M. Ba. acknowledges the Italian National Biodiversity Future Center (NBFC): National Recovery and Resilience Plan (NRRP), Mission 4 Component 2 Investment 1.4 of the Italian Ministry of University and Research; funded by the European Union – NextGenerationEU (Project code CN_00000033). M.W. acknowledges financial support by the German climate modeling project PalMod supported by the German Federal Ministry of Education and Research (BMBF) as a Research for Sustainability initiative (FONA) (grant nos. 01LP1920B and 01LP1917D). D. S. acknowledges financial support by the RRI "Tackling Global Change" from University of Bordeaux and by the INSU/LEFE DECORATING and the UKRI DECADAL projects.



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
