# Peer review of "Climate tipping point interactions and cascades: A review"

_EGUsphere, 2023_

## Author Comment (AC1)

**Dear Reviewers, dear Editor,**

**We are very grateful for the positive evaluation of our manuscript and the detailed feedback by the four reviewers, which encourages us to strongly revise our manuscript regarding the following important points:**

1. **We will introduce IPCC language regarding uncertainty notions of tipping element linkages.**
2. **We will add an overview matrix of all interactions to improve readability of key paper parts.**
3. **We will add a new figure on the relevant interactions during the Eocene-Oligocene Transition.**
4. **We will add an interpretation section of our results (for policy makers and the interested public). We conclude that tipping elements should not only be studied in isolation, but more emphasis has to be put on potential interactions, in particular given that the majority of links between tipping elements appear to be destabilizing.**
5. **Based on the reviewer comments, we aim to move section 4 on *Arctic sea ice →  coastal permafrost* to the supplementary information but keep and substantiate key parts of this section.**

**Please find below a detailed point-by-point response to the comments of the reviewers.**

**On behalf of all coauthors,**
**Nico Wunderling, Anna von der Heydt**

**Reviewer #1 (Steven Lade)**

COMMENT 1: This article is a thorough, well-written and timely review of interactions between climate tipping points.

**We are grateful for this positive evaluation of our manuscript.**

COMMENT 2: Section 2 is the key part of the paper. I am not sufficiently familiar with recent research on tipping point interactions to evaluate the statements made there. However all sound broadly plausible. I focus my attention instead on the other Sections.

Abstract

- A review paper should deliver findings or "insights". No insights are present in the abstract. It is all about the research gap and data sources. What are the key results from this paper? That tipping cascades are likely/unlikely to occur within the next couple of centuries?That interactions are mostly stabilising / destabilising? Rather than saying that you "identify crucial knowledge gaps", what are examples of the most urgent of those gaps?

**The reviewer is correct that we omitted an insight statement due to the (still) large uncertainties in tipping point interactions. However, based on the reviewer comment, we agree that a clear result/insight statement should be stated in the abstract and discussion of the paper.**

**We conclude that most of the interactions are indeed destabilizing (while uncertainties are large). Therefore, we conclude that tipping elements should not only be studied in isolation, but more emphasis has to be put on potential interactions. Further, tipping cascades cannot be ruled out when tipping thresholds of first tipping elements are transgressed through ongoing global warming. We will designed a new matrix (as a new figure) that summarizes Table 1 visually:**

[Figure]

**Fig. R1:** *Matrix of links between elements (tipping elements and other nonlinear components). Columns denote the element from which the interaction originates, rows denote the tipping element which is affected by the interaction. We separate three different types of effects. A stabilizing link(blue box), a destabilizing link (red box) and an unclear or competing link (gray box). White boxes denote no (or an unknown) link. Based on the recent literature, the strengths of the links are grouped into four groups: Strong (S), Moderate (M), Weak (W), and Unclear (U) if a strength estimate is lacking. Abbreviations of the elements stand for: GIS = Greenland Ice Sheet, WAIS = West Antarctic Ice Sheet, AMOC = Atlantic Meridional Overturning Circulation, ASI = Arctic Sea-Ice, AMAZ = Amazon rainforest, ENSO = El Niño-Southern Oscillation, Coral = Coral reefs, ISM = Indian Summer Monsoon, WAM = West African Monsoon, PERM = Permafrost*

Given the current state of research, it is yet unclear at which point in time (next decades/centuries/etc.) cascading transitions can be expected. Therefore, crucial knowledge gaps in interactions between tipping elements should be eliminated by four strategies (which we will outline in the new version of the manuscript):

I.    Observations-based approaches, e.g. causal inference

II.    **Earth system modeling, e.g. dedicated GCM/EMIC simulations**
III.   **Computational approaches, e.g. through neat propagation of uncertainties in ensemble methods (e.g. latin hypercube sampling, Monte Carlo methods)**
IV.    **Experts, e.g. an expert elicitation**

COMMENT 3: Section 1 introduces the study and makes some key definitions. These definitions could be tightened. For example:

- Lines 23-24 need work. A non-linear response (of output to input) is far from sufficient for a tipping point (e.g. the response could be x^2 rather than x). It is not clear what reorganisation means in this context. The reference cited is Armstrong McKay et al. but this is not the definition that those authors use.

**We thank the reviewer for pointing this out. We aim to include all elements that have a nonlinear response to forcing.**

**The core definition of a tipping point that we use originates from Levermann et al. (2012, Climatic Change, page 846): A reorganization of a tipping element means a qualitative change in the tipping element (e.g. from an ice-covered to an ice-free state in case of the Greenland Ice Sheet). During this reorganization process, the self-amplifying feedbacks (such as the ice-albedo or the melt-elevation) dominate the dynamics of the tipping elements. However, our definition also includes more regional bistabilities between savanna and forest vegetation in the Amazon region. In addition, we also consider elements that can show nonlinear behavior but it is speculated whether they should be considered tipping elements (e.g. El Niño Southern Oscillation: ENSO, Arctic sea ice, and Indian summer monsoon in this manuscript). These entities are important due to their connections to tipping elements and for Earth system stability. We therefore do not restrict ourselves to the most plausible tipping elements in this review, but also include nonlinear components like Arctic sea ice, ENSO and monsoon systems that can act as mediators of tipping events in the Earth system.**

**We will rewrite our definition section in the manuscript accordingly.**

**Reference:**

**Levermann, A., Bamber, J. L., Drijfhout, S., Ganopolski, A., Haeberli, W., Harris, N. R., Huss, M., Krüger, K., Lenton, T. M., Lindsay, R. W., et al.: Potential climatic transitions with profound impact on Europe: Review of the current state of six 'tipping elements of the climate system', Climatic Change, 110, 845–878, 2012.**

COMMENT 4:

- What exactly do "nonlinear behaviour" at line 27 and "nonlinear component" at line 49 mean? The relationship between driver and response is nonlinear? There is behaviour that matches that from nonlinear differential equations?

- Line 28: "we also consider elements that can show nonlinear behavior without being tipping elements on their own." In Fig 1, all elements are listed as either "tipping elements" or "speculative tipping elements" (which is not defined, but I presume means could be confirmed as a tipping element with more data). There is no category "nonlinear but non-tipping element" (to use the authors' language). Which are the "elements that can show nonlinear behavior without being tipping elements on their own"?

**These two points are important to clarify. We meant that we include all entities whose response to forcing is nonlinear and where the balance of feedbacks changes in a way that creates a higher sensitivity to forcing. Thus, not all considered entities need to be a tipping element in the strict sense themselves (see Armstrong McKay et al., 2022, Science, Main and SI tables). Those entities are the El Niño Southern Oscillation, Arctic sea ice, and Indian Summer Monsoon, which we included because of their importance to the worldwide interaction network from/to other tipping elements. We will reformulate the according sentences in the revised version of the manuscript.**

**Further, we will now omit the notion of *speculative* tipping elements and call them *nonlinear Earth system components*.**

COMMENT 5:

- Line 63 says that Arctic sea ice has almost a linear response to CO2 forcing, but then in line 81 say that "sea ice switches" are a "nonlinear element". Is this the same sea ice phenomenon?

**That is a very good point that the reviewer raises. The Arctic summer sea ice reacts nearly linearly to CO2 forcing (Notz & Stroeve, 2016) and can therefore not be considered a tipping element. The Arctic winter sea ice may be a tipping element that reacts nonlinearly to CO2 forcing (e.g. Hankel & Tziperman, 2023). However, for this study both parts of the Arctic sea ice (summer and winter) are important because of their interactions with and impact on the AMOC. Therefore, we will decide to take the Arctic sea ice as a single node in our network comprising the summer and winter sea ice. We will clarify this in the new manuscript.**

**References:**

**Hankel, C., E. Tziperman (2023). An approach for projecting the timing of abrupt winter Arctic sea ice loss. *Nonlinear Processes in Geophysics*, https://doi.org/10.5194/npg-30-299-2023.**

**Notz, D., J. Stroeve (2016). Observed Arctic sea-ice loss directly follows anthropogenic CO2 emission. *Science*, https://doi.org/10.1126/science.aag2345.**

COMMENT 5: Section 2.8

- Line 383: The finding by Wang et al. that "a tipping point cascade with large temperature feedbacks over the next couple of centuries remains unlikely" is cited uncritically. However, many interactions in Table 1 / Fig 2 occur on centuries or less. How do you reconcile these two viewpoints? Are there likely to be cascades within the next couple centuries that have significant impacts on climate and/or people?

Wang et al. (2023) state that tipping cascades with large temperature feedbacks leading to a hothouse climate state remain unlikely and are secondary to human emission trajectories. What we mean by this sentence is that the temperature feedbacks (on GMT) from tipping events or cascades are lower than the warming from direct anthropogenic emissions and are likely not leading to a runaway climate state. This statement is valid from our point of view given the current state.

Having said this, we have not commented on the likelihood of cascades (on centennial timescales and less) and their impact on climate and people in the old version of our manuscript, which we will do in the new version of the paper. We will add the following points to this manuscript:

1. Many of the known interactions are destabilizing (see Fig. R1 above) and the core tipping elements with the lowest thresholds are the large ice sheets on Greenland and West Antarctica. Therefore, they are usually the initiators of tipping cascades (Wunderling et a., 2021, ESD). However, those are also the tipping elements with the longest tipping time (several centuries up to millennia). Thus, if their tipping points are only transgressed for a limited amount of time (overshoot), also cascading tipping risks are well reduced (see supplementary Fig. 1a,b,c in Wunderling et al., 2023, Nat. Clim. Change).

2. But if global temperatures reach global warming levels beyond 2.0°C (or stay between 1.5-2.0°C on a centennial time scale), also more and more fast tipping elements like the AMOC or the Amazon rainforest would be at risk of tipping and could then potentially start a cascading transition on a much faster timescale (see supplementary Fig. 1d,e in Wunderling et al., 2023, Nat. Clim. Change).

Still, much uncertainty remains and most studies as of now rely on conceptual models rather than detailed process-based Earth system models. We have combined this with a later comment (COMMENT 9) and will add these points to our discussion in the new manuscript.

Reference:

Wunderling, N., Winkelmann, R., Rockström, J., Loriani, S., Armstrong McKay, D.I., Ritchie, P.D., Sakschewski, B. and Donges, J.F., 2023. Global warming overshoots increase risks of climate tipping cascades in a network model. *Nature Climate Change*, *13*(1), pp.75-82.

COMMENT 7: Sections 3 and 4

- These sections are interesting but introduced poorly. What purpose does introducing these examples serve? I see it as mainly establishing that tipping cascades are plausible, but this point is not made.

**We agree with the reviewer. The purpose of these two sections is to outline that tipping cascades are plausible and have been observed in the past (section 3) and in recent times (section 4, which we will move to SI also in line with the other reviewers). Further, also in line with the other reviewers comments, we will remove the word *archetypal* from our paper.**

COMMENT 8:

- In what sense is the speculated cascade in section 4 an "archetypal example" (see title of Section 4)? It's a speculation; in my opinion certainly not an archetype.

**We agree with the reviewer and will remove *archetypal* throughout the manuscript.**

COMMENT 9: Section 6

- Section 6 is very good at outlining possible directions for future research. I would like to see more discussion of the implications for non-researchers. Does this study show policymakers should be worried about tipping point interactions?

**These points were indeed lacking from the manuscript due to our cautiousness but we agree with the reviewer that a careful assessment of cascading tipping risks are needed, and will be added to the new manuscript.**

COMMENT 10: Comments on methods

- The paper lacks description of the literature review method. How do you know you haven't missed key literature on interactions? The method doesn't necessarily have to be a systematic review — it may be possible to argue that snowball is also acceptable.

**For each of the sections in this work, the lead-authors assigned a diverse group of long-standing experts in the field (who gratefully agreed to contribute and themselves added valuable advice on further expertise). As such, we are confident that we captured the most important parts of the literature on interacting tipping elements and cascading transitions. We will mention this in the new version of the manuscript.**

COMMENT 11:

- Section 1 mentions that any "linkage between tipping elements" is called a tipping interaction, with one element tipping causing another element tipping the "extreme case". Section 6 acknowledges that tipping interactions may be non-stationary, that is,

the interaction may change between the cases described above. Section 3, however, mixes these cases. For example, GIS->AMOC describes effects of any GIS melting on the AMOC. While 2.2.2 describes the effects of AMOC tipping. I suggest to either make the choice to not distinguish this difference more clear earlier on, or to include in e.g. Table 1 a summary of what type of interaction (e.g. pre or post tipping) was used in each assessment.

**That is a good suggestion. We do not want to distinguish between different types of interactions, also because this wouldn't be possible across the different studies. We will state that more clearly in the manuscript now.**

COMMENT 11: Minor Comments

- Line 194 - so winter Arctic sea ice could be considered a tipping element even though summer is not? If so it would be helpful to state more directly.

**That is indeed the case. Arctic winter sea ice may be considered a tipping element, while Arctic summer sea ice is not (see Armstrong McKay et al., 2022, Science). We will make that clearer in the new version of the manuscript.**

COMMENT 12:

- I don't understand the title of section 2.2.1. Do you perhaps mean "differentiating" or "distinguishing"?

**Thank you. There is indeed something wrong. We will rename this subsection to *Effects of disintegrating ice sheets on AMOC.***

**Reviewer #2 (anonymous)**

COMMENT 1: This is a nice, generally well and clearly written review that concentrates on interactions between tipping elements and the potential for cascades or stabilisations. Section 2 and table 1 are particularly useful. Some of the language is verging towards hyperbole in places however and could be more precise and scientific, particularly in section 1.

**We are thankful for this positive evaluation of our review and will follow the recommendation of this (and other) reviewers in order to avoid hyperbole language. With the help of the reviewers comments, we are now confident that the language changes lead to a more concise and scientific language (particularly in section 1).**

COMMENT 2: I have a few comments and questions listed below:

Line 21: 'These processes are at the heart of tipping behavior in the climate system and were found in numerous subsystems of the climate system.' Is this actually true? There are plenty of hypothesized tipping points in simple models and some GCMs but in the actual climate system? This sentence needs further qualification, it currently reads as if there is tipping all over the real world.

**We agree with the reviewer that the impression of this sentence is not entirely what we intended. As the reviewer said there are several hypothesized tipping elements found in simple models, EMICs and some GCMs, but also in paleo reconstructions (e.g. for the AMOC see our chapter 3). But since this is not the purpose of this manuscript to outline evidence for single tipping elements (an excellent review on this is Armstrong McKay et al., 2022, Science), we agree with the reviewer and will remove this sentence from our manuscript.**

COMMENT 3: Line 70: 'While most TEs that have been proposed so far are clearly regional (with some being large scale), there are significant knowledge gaps with respect to their tipping probability, impact estimates, time scales, as well as their interactions.' Why are there significant knowledge gaps? It would be good to outline the main reasons for this after this sentence. This would be very useful information for the reader. Is the main reason the lack of evidence of tipping in observations?

**This is a very good point that we will add after this sentence to the new manuscript. The main reasons for the knowledge gaps are manifold. Among others: (i) Experiments from comprehensive and process-based models are sparse (e.g. for the so-called *TIPMIP*-project many more of those experiments/simulations are planned in a systematic way), (ii) Luckily, we are lacking recent observations of tipping events (but if we had them, uncertainties were reducible), (iii) Paleoclimate observations are very helpful but still sparse in availability (some examples are listed in our section 3).**

COMMENT 4: Line 358: 'Insofar the hydrological cycle due to Permafrost changes may have far-reaching impacts.' Sentence does not read well. Change to something like 'Permafrost changes may impact the hydrological cycle with far-reaching impacts.'

**We will change the according sentence following the reviewers recommendation.**

COMMENT 5: Line 360: Title could be misleading. Suggest change to 'Interactions between multiple tipping elements and planetary **scale** cascades.' or something similar  (global scale cascades etc)

**We agree and will adopt the new section title.**

COMMENT 6: Title of section 3: I suggest you exchange 'Archetypal' with 'Possible'  or just remove archetypal – Archetypal suggests that these are well accepted examples which, later in 3.1 are acknowledged to only be 'possible'.

**In line and agreeing with this and the other reviewers, we will remove *archetypal* from our manuscript.**

COMMENT 7: Section 3.1: While the example is interesting from an Earth history point of view, is it really relevant to the subject of the review i.e. abrupt changes causing other abrupt changes? I'm fine with the inclusion of this material but this feels more like 'discussion/outlook' content

**We thank the reviewer for this impression and agree that this example from the more distant past may appear more speculative since direct and ample observational evidence is not easy to get for times that are so far in the past. However, we are strongly convinced that the Eocene-Oligocene Transition provides a good possible example for a past transition that involves several elements (Antarctic Ice sheet, monsoon systems, polar sea ice, etc.) and also includes a temporal component (of which element changed first and which later). Therefore, we prefer to keep it and hope the reviewer agrees with its inclusion.**

COMMENT 8: Section 3.2, line 444: …' with temperature increase in Greenland by 10-14°C over a few years; Andersen et al. (2004)'. Over a few years?! Is this true? That is incredibly fast. I read the citation (Andersen, 2004) with this statement and found no evidence of this claim. Please look at this again.

**We thank the reviewer for catching this. It has been found that NGRIP ice core data reveal that polar atmospheric circulation can shift in 1 to 3 years, resulting in decadal- to centennial-scale changes from cold stadials to warm interstadials/interglacials associated with large Greenland temperature changes of 10 K within several decades or less (Wolff et al., 2010; Steffensen et al., 2008; Landeis et al., 2005; Severinghaus and Brook, 1999). We will expand our manuscript accordingly.**

**References:**

- **Wolff, E.W., Chappellaz, J., Blunier, T., Rasmussen, S.O. and Svensson, A., 2010. Millennial-scale variability during the last glacial: The ice core record. *Quaternary Science Reviews*, *29*(21-22), pp.2828-2838.**
- **Steffensen, J.P., Andersen, K.K., Bigler, M., Clausen, H.B., Dahl-Jensen, D., Fischer, H., Goto-Azuma, K., Hansson, M., Johnsen, S.J., Jouzel, J. and Masson-Delmotte, V., 2008. High-resolution Greenland ice core data show abrupt climate change happens in few years. *Science*, *321*(5889), pp.680-684.**
- **Landais, A., Jouzel, J., Masson-Delmotte, V. and Caillon, N., 2005. Large temperature variations over rapid climatic events in Greenland: a method based on air isotopic measurements. *Comptes Rendus Geoscience*, *337*(10-11), pp.947-956.**
- **Severinghaus, J.P. and Brook, E.J., 1999. Abrupt climate change at the end of the last glacial period inferred from trapped air in polar ice. *Science*, *286*(5441), pp.930-934.**

COMMENT 9: Same thing again, title of section 4: Remove 'Archetypal'

**In line and agreeing with this and the other reviewers, we will remove *archetypal* from our manuscript.**

COMMENT 10: What is the difference between a tipping element and a nonlinear climate component? It would be good to define their use somewhere in the manuscript, ideally in the introduction.

**The reviewer is right that we need to more clearly distinguish tipping elements from nonlinear climate components. Some tipping elements are not tipping elements in the strict sense (e.g. ENSO, Arctic sea ice, Indian summer monsoon) but are still important Earth system components because they either interact with other tipping elements or mediate tipping effects further. We will note this in the introduction and will adapt our Figs. 1 and 2 accordingly.**

COMMENT 11: Line 547: 'Thus, (4) there is a potential cascading risk of large carbon releases to the ocean and atmosphere due to the coastal collapse. At the same time, coastal ecosystems would be impacted through increased nutrients and other terrigenous matter fluxes as well as local communities and economies (fisheries and infrastructure collapse).' I think more careful qualification is needed here. This is essentially a regional (Arctic) effect rather than a global one?. I suggest adding this qualification i.e. : 'Thus, (4) there is a potential cascading risk of large carbon releases to the **Arctic** ocean due to the coastal collapse. At the same time, **Arctic** coastal ecosystems would be impacted through increased nutrients and other terrigenous

matter fluxes as well as local communities and economies (fisheries and infrastructure collapse).'

**We agree with the reviewer that this is a regional effect and not a global effect and follow the reviewers recommendation. Note that we will move section 4 to the supplementary information in line with all reviewer comments, will modify the statement and add relevant references.**

COMMENT 12: Section 5: Are there any examples of cascades of tipping points in reasonably realistic models (EMIC, GCMs) under realistic forcing? These examples would be good to note here for the reader. If they have not yet been found, this would also be good to note in this section.

**It is important to know that we are not aware of fully coupled GCM or EMIC simulations of cascades of tipping points. Therefore, we think that it is important to note that such complex process-based models with sufficient integration of tipping elements (e.g. in EMICs, or GCMs) are only now starting to become available (for instance when ice-sheet models are dynamically integrated in global climate models, or when new versions of EMICs are getting developed). This will open up new possibilities of simulation cascading transitions. We will add a statement along those lines to the manuscript.**

**Reviewer #3 (B. van der Bolt)**

COMMENT 1: This is a well-written, interesting and thorough review of the interactions between tipping elements and the potential for cascades.

**We thank the reviewer for this positive assessment.**

COMMENT 2: What was unclear to me, however, was the relation between section 3 and 2. In addition, the purpose of section 4 is not clear at all, and the link between section 5 and the first sections is also not clear. To me, it seems that your article is split up in the following parts: introduction of the relevant concepts, overview of the possible interactions between climate tipping elements, evidence for tipping cascades from paleoclimatic data, and model approaches to study tipping cascades.

But how these parts lead up to an overall conclusion or overview, remains unclear. The article could be improved by better explaining the argumentation/story line of the paper and improving the connection between the different sections. In addition, I have some small comments which can be found below.

**We agree with the reviewer that the line of thought of the paper must be more readily understandable. Therefore, we follow the recommendation of this reviewer (and earlier reviewers) and will introduce our line of thought more clearly. Our plan for this article was as follows:**
**We indeed aimed to summarize the current knowledge on tipping interactions (and tipping cascades). Therefore, after laying out the concepts in the introduction (section 1), we started with reviewing the literature on pairwise interactions (section 2). Subsequently, we aimed at establishing that tipping cascades are plausible from a paleoclimate point of view (section 3) and a more recent point of view (old section 4, which we will now move to SI as we agree with the reviewer comment). This leads then to the question how to model tipping cascades if they are plausible (section 4). Section 5 then concludes our findings in two ways (i) should we worry about tipping cascades?, (ii) what needs to be done to improve our understanding of tipping cascades and associated risks?**

**We agree that much of this argumentation was missing from the original manuscript and will therefore adapt the manuscript in several paragraphs.**

*Please not that I do not have sufficient expertise on the topic of tipping point cascades and all the different climate elements to judge whether the main part of the paper includes all the relevant literature and if the information in that section is correct. I therefore focus in that part of the paper on the readability.*

**Section 1:**

COMMENT 3: Line 22: the words 'were found in numerous subsystems of the climate system' implies to me that they are observed in these subsystems, while the evidence for this is more nuanced. This part would be me accurate if you mention that there are indications for tipping point behaviour in these subsystems, based on model observation and paleoclimate datasets.

**We agree. In line with an earlier comment (Comment #2 of reviewer2), and will remove this sentence.**

COMMENT 4: Line 47: what do you mean with 'non-linear components' and how do they differ from the tipping elements you introduced in the section before?

**We agree and also in line with earlier reviews, we will now explain what we mean by nonlinear component early on in the manuscript. A nonlinear component for us is a speculative tipping element such as ENSO or the Arctic sea ice, which is not a tipping element in the strict sense (see Armstrong McKay et al., 2022, Science). Nevertheless, these elements are important for interactions with other tipping elements and thus for Earth system stability in general. Therefore, we think these elements are important to include (and agree that our earlier notion was confusing).**

COMMENT 5: Lines 50-52: Why do you introduce the different types of mathematical bifurcations in relation to the tipping cascades, and not in section 1.1 when you describe the definition of tipping elements?

**We agree with the reviewer and will move this section.**

COMMENT 6: Lines 63-64: The readability of this sentence would be improved if you cut it up into two sentences.

**We agree and will cut the sentence into two.**

**Section 2:**

COMMENT 7: Figure 1: The coral reef circles have – I think these are islands – in them, which makes the legenda a bit confusing. The figure would be more clear if you position these circles in a location where there is no island underneath (I do really like the coral reef icon in this figure).

**We thank the reviewer for this positive assessment, and will adapt the coral reef circles to positions without islands.**

COMMENT 8: Line 305: 'decreasing resilience' in this sentence now seems like one of the causes for a decrease in recovery time, while  the decrease of resilience is caused by the changes mentioned earlier in the sentence (warming temperatures… ocean acidification). This sentence could be formulated more clearly to show that the resilience decreases as a consequence of these changes.

**Yes, this is correct and the decrease of resilience could be read as an early warning signal, which isn't meant (as the reviewer states). To avoid confusion, we think the easiest solution is to remove *decreasing resilience* because the sentence also fully works without.**

**Section 3**

COMMENT 9: Title section 3: I suggest you change this title to 'Possible examples of interactions…"

**Also in line with earlier reviewer comments, we will change the title of this section accordingly.**

COMMENT 10: Line 440: I suggest you change 'chapter' to 'section' and include an introductory sentence to the other sections of section 3, or remove this one for consistency.

**We agree and will remove this sentence.**

**Section 4**

COMMENT 11: The purpose of section 4 is not clear to me. How does it relate to the previous sections and what does it add to the possible examples from the paleoclimate? To me, this example seems more like one of the possible cascades as introduced in section 2, than an archetypical example of nonlinear climate component interactions.

**We will move a strongly shortened version of this section to a new section "Arctic sea ice → Greenland Ice Sheet and permafrost". We will substantiate this section with additional references and add the (highly uncertain) links from the Arctic sea ice to the Greenland Ice Sheet and the permafrost to our figures and table.**

**Reviewer #4 (anonymous)**

COMMENT 1: The authors present a valuable review on the emerging aspect of Tipping Points (TPs) of TP interactions and TP cascades. Possible interactions are put into a spatial and temporal context, and the potential stabilizing and destabilizing processes are described. A specific example from the past is presented and a sequence of processes in the atmosphere, ocean and on land is sketched. The authors also offer a present-day example: a tipping point in Arctic ice loss triggers a tipping point in coastal retreat.

**We are thankful for this positive evaluation of our review paper.**

COMMENT 2: While the paper represents a valuable resource for scholars to inform themselves about this emerging topic, the considerations are speculative in many instances and important caveats and flags are missing. The authors should thoroughly parse the text for such speculations and hypotheses. A more nuanced text and a more formalized way of addressing uncertainties and limits in scientific understanding and knowledge should be the ambition for a revised version of this paper, before it is acceptable for publication.

**We thank the reviewer for their read through our manuscript and agree that a more nuanced text in some parts of the manuscript is warranted, as the other reviewers also suggested. For instance, we will remove the word *archetypal* from the section headings 3 and 4 (replacing it with possible), and among others will change the manuscript in several places.**

**We are looking forward to further feedback and are convinced that our manuscript will improve its language substantially.**

COMMENT 3: Specific Comments:

1) Table 1 provides a systematic description of the different links between two TPs depicted in Figure 1. Column 3 is key in stating the stabilizing or destabilizing nature of the link. However, the column sometimes also provides an uncertainty qualification, such as "highly uncertain". This is not done in this table in a systematic way. Here it is recommended that the IPCC approach on uncertainty language is adopted (Mastrandrea, M.D., et al., The IPCC AR5 guidance note on consistent treatment of uncertainties: a common approach across the working groups, Clim. Change, 108, 675-691, 2011.). Where possible, also the concept of risk should be applied (Reisinger et al., The Concept of Risk in the IPCC Sixth Assessment Report: A Summary of Cross-Working Group Discussions. IPCC, 2020). This would increase the value of this review, would make it more consistent with the broader, comprehensive assessments which also address TPs in several places, and provide consistency in this very complex topic.

2) In order to facilitate the implementation of 1) above, it is suggested that the authors add two additional columns in Table 1 before the columns collecting the references: (i) a column stating the uncertainty and the level of confidence, and (ii) a column stating the risk associated with the realization of this particular "cascade step" (from TP1 to TP2).

We thank the reviewer for this suggestion that indeed makes our assessment better comparable to established IPCC notions. Therefore, we will apply the following major changes to our manuscript.

Regarding uncertainty: We agree that adding levels of confidence to Tab. 1 increases the comparability and current state of research, which we will add as a **new column to Tab. 1** in the manuscript. It is, however, important to note that uncertainties in tipping points are large, and therefore uncertainties in tipping cascades are even larger. Most of the research on cascading transitions in climate has been performed using conceptual models, while more complex process-based models like EMICs or GCMs only start to become available (we will discuss that more thoroughly in the new manuscript). Therefore, our current assessment needs to remain qualitative using the notion of Fig. 2 in Mastrandrea et al., 2011.

Regarding risk: We also agree that another column (in Tab. 1) stating the risk associated with the realization of a particular "cascade step" would increase the impact of this work. Unfortunately, we feel that adding a credible level of risk is beyond the scope of the current literature. What we can say is the following: while confirmation or rejection through future research is necessary, it cannot be ruled out that interactions between climate tipping elements destabilize the Earth system in addition to climate change effects on individual tipping elements. Therefore, also tipping cascades cannot be ruled out when tipping thresholds of first tipping elements are transgressed through ongoing global warming (we will discuss this in the new manuscript). Providing a risk assessment will only be possible (i) for a given level of global warming or a given temperature trajectory, and (ii) with dedicated studies that assess a particular interaction because more complex process-based models like EMICs or GCMs only start to become available (this will also be added to the new manuscript version). However, we consent that future updates on this review should aim at such risk assessments.

COMMENT 5: 3) In the present-day example depicted in Fig. 4. the TP 2 "Coastal retreat" is illustrated (more on this below). This is missing from the map of Fig. 1 and the spatio-temporal diagram of Fig. 2. Since this is one of only two examples illustrated in a Figure, it would be important to include this in the two first figures.

4) Whether "coast retreat" is a TP or not is hard to qualify. Currently, the case for a TP is not sufficiently well made or convincing.

Following this reviewer (as well as other reviewer) recommendation, we will move a strongly shortened version of this section to a new section "Arctic sea ice → Greenland Ice Sheet and permafrost". We will substantiate this section with additional references and will add the (highly uncertain) links from the Arctic sea ice to the Greenland Ice Sheet and the permafrost to our figures and table. The rest of the old section 4 will be placed in the supplementary information due to its more speculative nature.

COMMENT 6: 5) Line 130: here some basic refs are missing, e.g.: Hu, A.X., et al., Energy balance in a warm world without the ocean conveyor belt and sea ice, Geophys. Res. Lett., 40, 6242-6246, 2013.

**We will add the additional reference to this part of the manuscript.**

COMMENT 7: 6) line 320: increased freshwater flux and AMOC reduction, along with a southward shift of the ITCZ appears many times in this manuscript. Please reduce redundant repetition.

**We will avoid repetition where possible in the new version of the manuscript.**

COMMENT 8: 7) lines 338-339: "relationship" is vague. Please clarify.

**This was indeed unclear in the old version of the manuscript. We will clarify the sentences accordingly in the revised version:**

**For example, while the linear effect of ENSO on the Indian Summer Monsoon rainfall has weakened, the effect of ENSO on the West African Monsoon rainfall has increased in recent decades. Both effects need to be further tested in paleoclimate reconstructions,**

COMMENT 9: 8) line 348: The high-latitude response of the hydrological cycle to an increase of GHGs is pretty robust (see IPCC, 2021). You presumably mean the regional response based on a catchment area perspective. Any ref for this?

**While the hydrological cycle in high latitudes is relatively robust (as the reviewer correctly noted), the representation of soil hydrology and its effect on Arctic and subarctic climate is not, as newest literature shows (de Vrese et al., 2023, The Cryosphere). We will clarify this in the new version of the manuscript.**

COMMENT 10: 9) line 372-378: this is an uncritical repetition of claims and speculations regarding the "hothouse" as a future possibility, promoted by Steffen et al. in 2018 and since then reiterated in several other papers (e.g. Kemp et al., 2022). The authors should provide the appropriate caveats, or else point to original research – not perspective papers or such – that simulate such effects. Repeating catch words without firm evidence should not feature in a review article and is not useful for the progress in this important topic.

**This is a valid point and it is important to underscore the hypothetical nature of this section. While the section after the mentioned lines already listed the caveats of the hothouse possibility, we will now strengthen the hypothetical nature and add additional references. We will also add that to our knowledge, no multi-millennial simulation with a model incorporating the elements and feedbacks described above has yet been conducted to test this scenario. This calls for experiments across the model hierarchy (we will add this discussion to the new version of the manuscript).**

COMMENT 11: 10) line 378: the breakup of stratocumulus decks, proposed by Schneider et al, is an interesting hypothesis valid in very limited areas in the subtropics. Furthermore, the tipping appears to occur at CO2 levels above 1200 ppm which is very high. Whether or not this purported instability, shown in a limited area LES model only, would translate into a large-scale effect is currently a speculation. This review should present this accordingly.

**The reviewer is correct and we will adapt our sentence on Schneider et al. (2019) accordingly by mentioning the preconditions for their strong temperature feedback: CO2 levels must be around 1200 ppm or larger.**

COMMENT 12: 11) Figure 3 illustrates a specific example in the paleoclimate record and panel b) provides some linkages. Some of these linkages are more robust, others are highly uncertain. It would be important to reflect this in, e.g., the line thickness of the arrows. Otherwise, the impression is given that all these links are equally well understood and quantified by, e.g., model simulations.

**The reviewer is right that some of the linkages are better constrained than others. We will replace arrows by dashed arrows to make clear which linkages are less constrained as compared to others. We will further add a figure on the Eocene-Oligocene transition, where all linkages are very uncertain (therefore all of them will be represented by dashed arrows).**

COMMENT 13: 12) line 614: "We conclude that tipping elements interact …" This appears as a very strong statement of fact. The authors provide no evidence for such robustness and hence a more cautious formulation should be chosen.

**We agree with the reviewer that a cautious notion of our results is necessary. At the same time, we were requested to more clearly lay out consequences of our results to policy makers (Comment 9 of reviewer #1: should policy makers be worried about tipping cascades?). Therefore, we will adapt our manuscript along the following lines:**

1. **Tipping elements interact across scales in space and time (see Fig. 1, Tab. 1).**
2. **The majority of tipping element linkages are destabilizing (13 destabilizing, 2 stabilizing, 4 unclear). Therefore, we conclude that tipping elements should not only be studied in isolation, but more emphasis has to be put on potential interactions, in particular given that the majority of links between tipping elements appear to be destabilizing. Thus, tipping cascades cannot be ruled out when tipping thresholds of first tipping elements are transgressed through ongoing global warming. However, decisive for the possibility of a tipping cascade is that a first tipping element shows signs of disintegration, i.e. tipping is initiated. Cryosphere tipping elements would be such candidates (Wunderling et al., 2021, ESD; Wunderling et al., 2023, Nat. Clim. Change).**
3. **Because studies as of now are based on conceptual models, we need detailed assessments of tipping elements and their interactions in process-based models**

**(EMICs or ideally GCMs). This is currently underway, for instance within the so-called TIPMIP endeavor.**

COMMENT 14: 13) line 634: "tremendous". Why should a non-quantitative study based on questions and conversations (expert elicitation) be of "tremendous value"? This is overselling this type of information gathering. Of more value would certainly be targeted simulations across a hierarchy of models, careful parameter and sensitivity studies, and large ensemble simulations. The justification of the "tremendous value" is that this would provide "direct expert input". Such "direct expert input" should be reflected in the authorship of, e.g., review papers.

**The reviewer is right that the word *tremendous* is misplaced here. We will remove it. We further think that an admixture of different methods is best to reduce uncertainties in tipping cascades. Probably the two best ways are (i) process-based modeling by GCMs or EMICs if models are available (agreeing with the reviewer) or (ii) direct observations or early warning signs of cascading transitions. As long as such methods are not available, also simpler models, ensemble approaches and expert knowledge are important to consider (this will be added to the manuscript).**

COMMENT 15: 14) line 676: This review paper ends by referencing a highly speculative piece which was a perspective paper, provocative and stimulating debate. However, it did not contain quantitative analysis or original research. Therefore, it should be considered an opinion piece and thus be treated in a scientific review accordingly. By citing such work in later scientific journals without the proper qualifiers lends undue support for what originally was a stimulating or provocative idea. Clearly, such work can and should be cited, but the context must be given appropriately.

**We agree with the reviewer and will omit the perspectives/opinion piece by Kemp et al in the last sentence and better integrate it with current literature.**

---

## Author Response (AR1)

Dear Reviewers, dear Editor,

We are very grateful for the positive evaluation of our manuscript and the detailed feedback by the four reviewers and the editor, which encouraged us to strongly revise our manuscript regarding the following four important points:

1. We have introduced IPCC language regarding uncertainty notions of tipping element linkages (see Tab. 1, new column).
2. We added an overview matrix (Fig. 3) of all interactions to improve readability of key paper parts. We also added a new figure on the relevant interactions during the Eocene-Oligocene Transition that was missing from the earlier version of the manuscript (Fig. 4).
3. Based on the reviewer comments, we have decided to move section 4 on *Arctic sea ice → coastal permafrost* to the supplementary information but kept and substantiated key parts of this section in the new section 2.3.2 (and added the linkages to Figs. 1-3).
4. We added an interpretation section of our results (for policy makers and the interested public). We conclude that tipping elements should not only be studied in isolation, but more emphasis has to be put on potential interactions, in particular given that the majority of links between tipping elements appear to be destabilizing.

Please find below a detailed point-by-point response to the comments of the reviewers. We attached the new version of our manuscript, and marked the changes in blue. We are grateful for this opportunity to substantially improve our manuscript and are looking forward to further feedback.

On behalf of all coauthors,
Nico Wunderling, Anna von der Heydt

**Reviewer #1 (Steven Lade)**

COMMENT 1: This article is a thorough, well-written and timely review of interactions between climate tipping points.

**We are grateful for this positive evaluation of our manuscript.**

COMMENT 2: Section 2 is the key part of the paper. I am not sufficiently familiar with recent research on tipping point interactions to evaluate the statements made there. However all sound broadly plausible. I focus my attention instead on the other Sections.

Abstract

- A review paper should deliver findings or "insights". No insights are present in the abstract. It is all about the research gap and data sources. What are the key results from this paper? That tipping cascades are likely/unlikely to occur within the next couple of centuries?That interactions are mostly stabilising / destabilising? Rather than saying that you "identify crucial knowledge gaps", what are examples of the most urgent of those gaps?

**The reviewer is correct that we omitted an insight statement due to the (still) large uncertainties in tipping point interactions. However, based on the reviewer comment, we agree that a clear result/insight statement should be stated in the abstract and discussion of the paper.**

**We conclude that most of the interactions are indeed destabilizing (while uncertainties are large). Therefore, we conclude that tipping elements should not only be studied in isolation, but more emphasis has to be put on potential interactions. Further, tipping cascades cannot be ruled out when tipping thresholds of first tipping elements are transgressed through ongoing global warming. We also designed a matrix (new figure 3 in the manuscript) that summarizes Table 1 visually:**

[Figure]

**Fig. R1:** *Matrix of links between elements (tipping elements and other nonlinear components) discussed in this chapter. Columns denote the element from which the interaction originates, rows denote the tipping element which is affected by the interaction. We separate three different types of effects. A stabilizing link(blue box), a destabilizing link (red box) and an unclear or competing link (gray box). White boxes denote no (or an unknown) link. Based on the recent literature, the strengths of the links are grouped into four groups: Strong (S), Moderate (M), Weak (W), and Unclear (U) if a strength estimate is lacking. Abbreviations of the elements stand for: GIS = Greenland Ice Sheet, WAIS = West Antarctic Ice Sheet, AMOC = Atlantic Meridional Overturning Circulation, ASI = Arctic Sea-Ice, AMAZ = Amazon rainforest, ENSO = El Niño-Southern Oscillation, Coral = Coral reefs, ISM = Indian Summer Monsoon, WAM = West African Monsoon, PERM = Permafrost*

Given the current state of research, it is yet unclear at which point in time (next decades/centuries/etc.) cascading transitions can be expected. Therefore, crucial knowledge gaps in interactions between tipping elements should be eliminated by four strategies (see ll **658-672**):

    I.    **Observations-based approaches, e.g. causal inference**

II.     **Earth system modeling, e.g. dedicated GCM/EMIC simulations**

III.    **Computational approaches, e.g. through neat propagation of uncertainties in ensemble methods (e.g. latin hypercube sampling, Monte Carlo methods)**

IV.    **Experts, e.g. an expert elicitation**

**We have rewritten the abstract (ll 9-14) and the discussion (ll 639-657).**

COMMENT 3: Section 1 introduces the study and makes some key definitions. These definitions could be tightened. For example:

- Lines 23-24 need work. A non-linear response (of output to input) is far from sufficient for a tipping point (e.g. the response could be $x^2$ rather than x). It is not clear what reorganisation means in this context. The reference cited is Armstrong McKay et al. but this is not the definition that those authors use.

**We thank the reviewer for pointing this out. We aim to include all elements that have a nonlinear response to forcing.**

**The core definition of a tipping point that we use originates from Levermann et al. (2012, Climatic Change, page 846): A reorganization of a tipping element means a qualitative change in the tipping element (e.g. from an ice-covered to an ice-free state in case of the Greenland Ice Sheet). During this reorganization process, the self-amplifying feedbacks (such as the ice-albedo or the melt-elevation) dominate the dynamics of the tipping elements. However, our definition also includes more regional bistabilities between savanna and forest vegetation in the Amazon region. In addition, we also consider elements that can show nonlinear behavior but it is speculated whether they should be considered tipping elements (e.g. El Niño Southern Oscillation: ENSO, Arctic sea ice, and Indian summer monsoon in this manuscript). These entities are important due to their connections to tipping elements and for Earth system stability. We therefore do not restrict ourselves to the most plausible tipping elements in this review, but also include nonlinear components like Arctic sea ice, ENSO and monsoon systems that can act as mediators of tipping events in the Earth system.**

**We have rewritten our definition section in the manuscript accordingly (see ll 26-37).**

**Reference:**

**Levermann, A., Bamber, J. L., Drijfhout, S., Ganopolski, A., Haeberli, W., Harris, N. R., Huss, M., Krüger, K., Lenton, T. M., Lindsay, R. W., et al.: Potential climatic transitions with profound impact on Europe: Review of the current state of six 'tipping elements of the climate system', Climatic Change, 110, 845–878, 2012.**

COMMENT 4:

- What exactly do "nonlinear behaviour" at line 27 and "nonlinear component" at line 49 mean? The relationship between driver and response is nonlinear? There is behaviour that matches that from nonlinear differential equations?
- Line 28: "we also consider elements that can show nonlinear behavior without being tipping elements on their own." In Fig 1, all elements are listed as either "tipping elements" or "speculative tipping elements" (which is not defined, but I presume means could be confirmed as a tipping element with more data). There is no category "nonlinear but non-tipping element" (to use the authors' language). Which are the "elements that can show nonlinear behavior without being tipping elements on their own"?

**These two points are important to clarify. We meant that we include all entities whose response to forcing is nonlinear and where the balance of feedbacks changes in a way that creates a higher sensitivity to forcing. Thus, not all considered entities need to be a tipping element in the strict sense themselves (see Armstrong McKay et al., 2022, Science, Main and SI tables). Those entities are the El Niño Southern Oscillation, Arctic sea ice, and Indian Summer Monsoon, which we included because of their importance to the worldwide interaction network from/to other tipping elements. We have reformulated the according sentences, see definition ll 26-37.**

**Further, we now omit the notion of *speculative* tipping elements and call them *nonlinear Earth system components* (see new Figs. 1 and 2).**

COMMENT 5:

- Line 63 says that Arctic sea ice has almost a linear response to CO2 forcing, but then in line 81 say that "sea ice switches" are a "nonlinear element". Is this the same sea ice phenomenon?

**That is a very good point that the reviewer raises. The Arctic summer sea ice reacts nearly linearly to CO2 forcing (Notz & Stroeve, 2016) and can therefore not be considered a tipping element. The Arctic winter sea ice may be a tipping element that reacts nonlinearly to CO2 forcing (e.g. Hankel & Tziperman, 2023). However, for this study both parts of the Arctic sea ice (summer and winter) are important because of their interactions with and impact on the AMOC. Therefore, we decided to take the Arctic sea ice as a single node in our network comprising the summer and winter sea ice. We clarified this in the manuscript (see ll 80-83).**

**References:**

**Hankel, C., E. Tziperman (2023). An approach for projecting the timing of abrupt winter Arctic sea ice loss. *Nonlinear Processes in Geophysics*, https://doi.org/10.5194/npg-30-299-2023.**

**Notz, D., J. Stroeve (2016). Observed Arctic sea-ice loss directly follows anthropogenic CO2 emission. *Science*, https://doi.org/10.1126/science.aag2345.**

COMMENT 5: Section 2.8

- Line 383: The finding by Wang et al. that "a tipping point cascade with large temperature feedbacks over the next couple of centuries remains unlikely" is cited uncritically. However, many interactions in Table 1 / Fig 2 occur on centuries or less. How do you reconcile these two viewpoints? Are there likely to be cascades within the next couple centuries that have significant impacts on climate and/or people?

**Wang et al. (2023) state that tipping cascades with large temperature feedbacks leading to a hothouse climate state remain unlikely and are secondary to human emission trajectories. What we mean by this sentence is that the temperature feedbacks (on GMT) from tipping events or cascades are lower than the warming from direct anthropogenic emissions and are likely not leading to a runaway climate state. This statement is valid from our point of view given the current state.**

**Having said this, we have not commented on the likelihood of cascades (on centennial timescales and less) and their impact on climate and people in the old version of our manuscript. We agree with the reviewer that we should comment on that. We added the following points in this manuscript regarding:**

1. **Many of the known interactions are destabilizing (see Fig. R1 above) and the core tipping elements with the lowest thresholds are the large ice sheets on Greenland and West Antarctica. Therefore, they are usually the initiators of tipping cascades (Wunderling et a., 2021, ESD). However, those are also the tipping elements with the longest tipping time (several centuries up to millennia). Thus, if their tipping points are only transgressed for a limited amount of time (overshoot), also cascading tipping risks are well reduced (see supplementary Fig. 1a,b,c in Wunderling et al., 2023, Nat. Clim. Change).**
2. **But if global temperatures reach global warming levels beyond 2.0°C (or stay between 1.5-2.0°C on a centennial time scale), also more and more fast tipping elements like the AMOC or the Amazon rainforest would be at risk of tipping and could then potentially start a cascading transition on a much faster timescale (see supplementary Fig. 1d,e in Wunderling et al., 2023, Nat. Clim. Change).**

**Still, much uncertainty remains and most studies as of now rely on conceptual models rather than detailed process-based Earth system models. We have combined this with a later comment (COMMENT 9) and added the points to our discussion, see ll 639-657.**

COMMENT 7: Sections 3 and 4

- These sections are interesting but introduced poorly. What purpose does introducing these examples serve? I see it as mainly establishing that tipping cascades are plausible, but this point is not made.

**We agree with the reviewer. The purpose of these two sections is to outline that tipping cascades are plausible and have been observed in the past (section 3) and in recent times (section 4, now moved to SI also in line with the other reviewers). Further, also in line with the other reviewers comments, we removed *archetypal* from our paper. Manuscript changes can be found in ll 98-103 and ll 436-438.**

COMMENT 8:

- In what sense is the speculated cascade in section 4 an "archetypal example" (see title of Section 4)? It's a speculation; in my opinion certainly not an archetype.

**We agree with the reviewer and removed *archetypal* throughout the manuscript.**

COMMENT 9: Section 6

- Section 6 is very good at outlining possible directions for future research. I would like to see more discussion of the implications for non-researchers. Does this study show policymakers should be worried about tipping point interactions?

**These points were indeed lacking from the manuscript due to our cautiousness but we agree with the reviewer that a careful assessment of cascading tipping risks are needed, see ll 639-657.**

COMMENT 10: Comments on methods

- The paper lacks description of the literature review method. How do you know you haven't missed key literature on interactions? The method doesn't necessarily have to be a systematic review — it may be possible to argue that snowball is also acceptable.

**For each of the sections in this work, the lead-authors assigned a diverse group of long-standing experts in the field (who gratefully agreed to contribute and themselves added valuable advice on further expertise). As such, we are confident that we captured the most important parts of the literature on interacting tipping elements and cascading transitions. We mention this now in ll 113-116.**

COMMENT 11:

- Section 1 mentions that any "linkage between tipping elements" is called a tipping interaction, with one element tipping causing another element tipping the "extreme case". Section 6 acknowledges that tipping interactions may be non-stationary, that is, the interaction may change between the cases described above. Section 3, however, mixes these cases. For example, GIS->AMOC describes effects of any GIS melting on the AMOC. While 2.2.2 describes the effects of AMOC tipping. I suggest to either make the choice to not distinguish this difference more clear earlier on, or to include in e.g. Table 1 a summary of what type of interaction (e.g. pre or post tipping) was used in each assessment.

**That is a good suggestion. We do not want to distinguish between different types of interactions, also because this wouldn't be possible across the different studies. We state that more clearly in the manuscript now (see ll 62-66).**

COMMENT 11: Minor Comments

- Line 194 - so winter Arctic sea ice could be considered a tipping element even though summer is not? If so it would be helpful to state more directly.

**That is indeed the case. Arctic winter sea ice may be considered a tipping element, while Arctic summer sea ice is not (see Armstrong McKay et al., 2022, Science). We have made that clearer in ll 26-37 and ll 80-83.**

COMMENT 12:

- I don't understand the title of section 2.2.1. Do you perhaps mean "differentiating" or "distinguishing"?

**Thank you. There is indeed something wrong. We have renamed this subsection to *Effects of disintegrating ice sheets on AMOC.***

**Reviewer #2 (anonymous)**

COMMENT 1: This is a nice, generally well and clearly written review that concentrates on interactions between tipping elements and the potential for cascades or stabilisations. Section 2 and table 1 are particularly useful. Some of the language is verging towards hyperbole in places however and could be more precise and scientific, particularly in section 1.

**We are thankful for this positive evaluation of our review and we have followed the recommendation of this (and other) reviewers in order to avoid hyperbole language. With the help of the reviewers comments, we are now confident that the language changes lead to a more concise and scientific language (particularly in section 1).**

COMMENT 2: I have a few comments and questions listed below:

Line 21: 'These processes are at the heart of tipping behavior in the climate system and were found in numerous subsystems of the climate system.' Is this actually true? There are plenty of hypothesized tipping points in simple models and some GCMs but in the actual climate system? This sentence needs further qualification, it currently reads as if there is tipping all over the real world.

**We agree with the reviewer that the impression of this sentence is not entirely what we intended. As the reviewer said there are several hypothesized tipping elements found in simple models, EMICs and some GCMs, but also in paleo reconstructions (e.g. for the AMOC see our chapter 3). But since this is not the purpose of this manuscript to outline evidence for single tipping elements (an excellent review on this is Armstrong McKay et al., 2022, Science), we agree with the reviewer and removed this sentence from our manuscript.**

COMMENT 3: Line 70: 'While most TEs that have been proposed so far are clearly regional (with some being large scale), there are significant knowledge gaps with respect to their tipping probability, impact estimates, time scales, as well as their interactions.' Why are there significant knowledge gaps? It would be good to outline the main reasons for this after this sentence. This would be very useful information for the reader. Is the main reason the lack of evidence of tipping in observations?

**This is a very good point that we added after this sentence (see new manuscript ll 89-94). The main reasons for the knowledge gaps are manifold. Among others: (i) Experiments from comprehensive and process-based models are sparse (e.g. for the so-called *TIPMIP*-project many more of those experiments/simulations are planned in a systematic way), (ii) Luckily, we are lacking recent observations of tipping events (but if we had them, uncertainties were reducible), (iii) Paleoclimate observations are very helpful but still sparse in availability (some examples are listed in our section 3).**

COMMENT 4: Line 358: 'Insofar the hydrological cycle due to Permafrost changes may have far-reaching impacts.' Sentence does not read well. Change to something like 'Permafrost changes may impact the hydrological cycle with far-reaching impacts.'

**We have changed the according sentence following the reviewers recommendation, see ll 398-399.**

COMMENT 5: Line 360: Title could be misleading. Suggest change to 'Interactions between multiple tipping elements and planetary **scale** cascades.' or something similar  (global scale cascades etc)

**We agree and adopted the new section title.**

COMMENT 6: Title of section 3: I suggest you exchange 'Archetypal' with 'Possible'  or just remove archetypal – Archetypal suggests that these are well accepted examples which, later in 3.1 are acknowledged to only be 'possible'.

**In line and agreeing with this and the other reviewers, we have removed *archetypal* from our manuscript.**

COMMENT 7: Section 3.1: While the example is interesting from an Earth history point of view, is it really relevant to the subject of the review i.e. abrupt changes causing other abrupt changes? I'm fine with the inclusion of this material but this feels more like 'discussion/outlook' content

**We thank the reviewer for this impression and agree that this example from the more distant past may appear more speculative since direct and ample observational evidence is not easy to get for times that are so far in the past. However, we are strongly convinced that the Eocene-Oligocene Transition provides a good possible example for a past transition that involves several elements (Antarctic Ice sheet, monsoon systems, polar sea ice, etc.) and also includes a temporal component (of which element changed first and which later). Therefore, we prefer to keep it and hope the reviewer agrees with its inclusion as subsection 3.1.**

COMMENT 8: Section 3.2, line 444: …' with temperature increase in Greenland by 10-14°C over a few years; Andersen et al. (2004)'. Over a few years?! Is this true? That is incredibly fast. I read the citation (Andersen, 2004) with this statement and found no evidence of this claim. Please look at this again.

**We thank the reviewer for catching this line. It has been found that NGRIP ice core data reveal that polar atmospheric circulation can shift in 1 to 3 years, resulting in decadal- to centennial-scale changes from cold stadials to warm interstadials/interglacials associated with large Greenland temperature changes of 10 K within several decades or less (Wolff et al., 2010; Steffensen et al., 2008; Landeis et al., 2005; Severinghaus and Brook, 1999). We have adapted our manuscript accordingly, see ll 488-490.**

**References:**

- **Wolff, E.W., Chappellaz, J., Blunier, T., Rasmussen, S.O. and Svensson, A., 2010. Millennial-scale variability during the last glacial: The ice core record.** *Quaternary Science Reviews*, *29*(21-22), pp.2828-2838.
- **Steffensen, J.P., Andersen, K.K., Bigler, M., Clausen, H.B., Dahl-Jensen, D., Fischer, H., Goto-Azuma, K., Hansson, M., Johnsen, S.J., Jouzel, J. and Masson-Delmotte, V., 2008. High-resolution Greenland ice core data show abrupt climate change happens in few years.** *Science*, *321*(5889), pp.680-684.
- **Landais, A., Jouzel, J., Masson-Delmotte, V. and Caillon, N., 2005. Large temperature variations over rapid climatic events in Greenland: a method based on air isotopic measurements.** *Comptes Rendus Geoscience*, *337*(10-11), pp.947-956.
- **Severinghaus, J.P. and Brook, E.J., 1999. Abrupt climate change at the end of the last glacial period inferred from trapped air in polar ice.** *Science*, *286*(5441), pp.930-934.

COMMENT 9: Same thing again, title of section 4: Remove 'Archetypal'

**In line and agreeing with this and the other reviewers, we have removed *archetypal* from our manuscript.**

COMMENT 10: What is the difference between a tipping element and a nonlinear climate component? It would be good to define their use somewhere in the manuscript, ideally in the introduction.

**The reviewer is right that we need to more clearly distinguish tipping elements from nonlinear climate components. Some tipping elements are not tipping elements in the strict sense (e.g. ENSO, Arctic sea ice, Indian summer monsoon) but are still important Earth system components because they either interact with other tipping elements or mediate tipping effects further. We noted this in the introduction, see ll 26-38 and adapted our Figs. 1 and 2 accordingly.**

COMMENT 11: Line 547: 'Thus, (4) there is a potential cascading risk of large carbon releases to the ocean and atmosphere due to the coastal collapse. At the same time, coastal ecosystems would be impacted through increased nutrients and other terrigenous matter fluxes as well as local communities and economies (fisheries and infrastructure collapse).' I think more careful qualification is needed here. This is essentially a regional (Arctic) effect rather than a global one?. I suggest adding this qualification i.e. : 'Thus, (4) there is a potential cascading risk of large carbon releases to the **Arctic** ocean due to the coastal collapse. At the same time, **Arctic** coastal ecosystems would be impacted through increased nutrients and other terrigenous

matter fluxes as well as local communities and economies (fisheries and infrastructure collapse).'

**We agree with the reviewer that this is a regional effect and not a global effect and follow the reviewers recommendation, see SI in ll 25-28. Note that we have moved section 4 to the supplementary information (SI) in line with all reviewer comments, modified the statement and added relevant references.**

COMMENT 12: Section 5: Are there any examples of cascades of tipping points in reasonably realistic models (EMIC, GCMs) under realistic forcing? These examples would be good to note here for the reader. If they have not yet been found, this would also be good to note in this section.

**It is important to know that we are not aware of fully coupled GCM or EMIC simulations of cascades of tipping points. Therefore, we think that it is important to note that such complex process-based models with sufficient integration of tipping elements (e.g. in EMICs, or GCMs) are only now starting to become available (for instance when ice-sheet models are dynamically integrated in global climate models, or when new versions of EMICs are getting developed). This will open up new possibilities of simulation cascading transitions. We added this statement to the manuscript, see ll 605-608.**

**Reviewer #3 (B. van der Bolt)**

COMMENT 1: This is a well-written, interesting and thorough review of the interactions between tipping elements and the potential for cascades.

**We thank the reviewer for this positive assessment.**

COMMENT 2: What was unclear to me, however, was the relation between section 3 and 2. In addition, the purpose of section 4 is not clear at all, and the link between section 5 and the first sections is also not clear. To me, it seems that your article is split up in the following parts: introduction of the relevant concepts, overview of the possible interactions between climate tipping elements, evidence for tipping cascades from paleoclimatic data, and model approaches to study tipping cascades.

But how these parts lead up to an overall conclusion or overview, remains unclear. The article could be improved by better explaining the argumentation/story line of the paper and improving the connection between the different sections. In addition, I have some small comments which can be found below.

**We agree with the reviewer that the line of thought of the paper must be more readily understandable. Therefore, we follow the recommendation of this reviewer (and earlier reviewers) and introduce our line of thought more clearly. Our plan for this article was as follows:**
**We indeed aimed to summarize the current knowledge on tipping interactions (and tipping cascades). Therefore, after laying out the concepts in the introduction (section 1), we started with reviewing the literature on pairwise interactions (section 2). Subsequently, we aimed at establishing that tipping cascades are plausible from a paleoclimate point of view (section 3) and a more recent point of view (old section 4, now moved to SI as we agree with the reviewer comment). This leads then to the question how to model tipping cascades if they are plausible (section 4). Section 5 then concludes our findings in two ways (i) should we worry about tipping cascades? (ll 639-657), (ii) what needs to be done to improve our understanding of tipping cascades and associated risks? (ll 658-672)**

**We agree that much of this argumentation was missing from the original manuscript and have therefore adapted the manuscript in several lines (see ll 98-103, ll 436-438, ll 639-657).**

*Please not that I do not have sufficient expertise on the topic of tipping point cascades and all the different climate elements to judge whether the main part of the paper includes all the*

*relevant literature and if the information in that section is correct. I therefore focus in that part of the paper on the readability.*

**Section 1:**

COMMENT 3: Line 22: the words 'were found in numerous subsystems of the climate system' implies to me that they are observed in these subsystems, while the evidence for this is more nuanced. This part would be me accurate if you mention that there are indications for tipping point behaviour in these subsystems, based on model observation and paleoclimate datasets.

**We agree. In line with an earlier comment (Comment #2 of reviewer2), we have decided to remove this sentence.**

COMMENT 4: Line 47: what do you mean with 'non-linear components' and how do they differ from the tipping elements you introduced in the section before?

**We agree and also in line with earlier reviews, we now explain what we mean by nonlinear component early on in the manuscript (see ll 26-38). A nonlinear component for us is a speculative tipping element such as ENSO or the Arctic sea ice, which is not a tipping element in the strict sense (see Armstrong McKay et al., 2022, Science). Nevertheless, these elements are important for interactions with other tipping elements and thus for Earth system stability in general. Therefore, we think these elements are important to include (and agree that our earlier notion was confusing).**

COMMENT 5: Lines 50-52: Why do you introduce the different types of mathematical bifurcations in relation to the tipping cascades, and not in section 1.1 when you describe the definition of tipping elements?

**We agree with the reviewer and moved this section, see ll 60-61.**

COMMENT 6: Lines 63-64: The readability of this sentence would be improved if you cut it up into two sentences.

**We agree and have cut the sentence into two.**

**Section 2:**

COMMENT 7: Figure 1: The coral reef circles have – I think these are islands – in them, which makes the legenda a bit confusing. The figure would be more clear if you position these circles in a location where there is no island underneath (I do really like the coral reef icon in this figure).

**We thank the reviewer for this positive assessment, and have adapted the coral reef circles to positions without islands.**

COMMENT 8: Line 305: 'decreasing resilience' in this sentence now seems like one of the causes for a decrease in recovery time, while the decrease of resilience is caused by the changes mentioned earlier in the sentence (warming temperatures… ocean acidification). This sentence could be formulated more clearly to show that the resilience decreases as a consequence of these changes.

**Yes, this is correct and the decrease of resilience could be read as an early warning signal, which isn't meant (as the reviewer states). To avoid confusion, we think the easiest solution is to remove *decreasing resilience* because the sentence also fully works without.**

**Section 3**

COMMENT 9: Title section 3: I suggest you change this title to 'Possible examples of interactions…"

**Also in line with earlier reviewer comments, we changed the title of this section accordingly.**

COMMENT 10: Line 440: I suggest you change 'chapter' to 'section' and include an introductory sentence to the other sections of section 3, or remove this one for consistency.

**We decided to remove this sentence.**

**Section 4**

COMMENT 11: The purpose of section 4 is not clear to me. How does it relate to the previous sections and what does it add to the possible examples from the paleoclimate? To me, this example seems more like one of the possible cascades as introduced in section 2, than an archetypical example of nonlinear climate component interactions.

**We moved a strongly shortened version of this section to our new section 2.3.2 (Arctic sea ice → Greenland Ice Sheet and permafrost). We substantiated this section with additional references and added the (highly uncertain) links from the Arctic sea ice to the Greenland Ice Sheet and the permafrost to our Figs. 1-3 and Tab. 1. The reworked text can be found in ll 217-229. The rest of the old section 4 has been placed in the supplementary information.**

**Reviewer #4 (anonymous)**

COMMENT 1: The authors present a valuable review on the emerging aspect of Tipping Points (TPs) of TP interactions and TP cascades. Possible interactions are put into a spatial and temporal context, and the potential stabilizing and destabilizing processes are described. A specific example from the past is presented and a sequence of processes in the atmosphere, ocean and on land is sketched. The authors also offer a present-day example: a tipping point in Arctic ice loss triggers a tipping point in coastal retreat.

**We are thankful for this positive evaluation of our review paper.**

COMMENT 2: While the paper represents a valuable resource for scholars to inform themselves about this emerging topic, the considerations are speculative in many instances and important caveats and flags are missing. The authors should thoroughly parse the text for such speculations and hypotheses. A more nuanced text and a more formalized way of addressing uncertainties and limits in scientific understanding and knowledge should be the ambition for a revised version of this paper, before it is acceptable for publication.

**We thank the reviewer for their read through our manuscript and, agree that a more nuanced text in some parts of the manuscript is warranted, as the other reviewers also suggested. For instance, we removed the word *archetypal* from the section headings 3 and 4 (replacing it with possible), and among others changed the following manuscript lines: ll 26-37, l 435, ll 605-608.**

**We are looking forward to further feedback and are convinced that our manuscript has now improved its language substantially.**

COMMENT 3: Specific Comments:

1) Table 1 provides a systematic description of the different links between two TPs depicted in Figure 1. Column 3 is key in stating the stabilizing or destabilizing nature of the link. However, the column sometimes also provides an uncertainty qualification, such as "highly uncertain". This is not done in this table in a systematic way. Here it is recommended that the IPCC approach on uncertainty language is adopted (Mastrandrea, M.D., et al., The IPCC AR5 guidance note on consistent treatment of uncertainties: a common approach across the working groups, Clim. Change, 108, 675-691, 2011.). Where possible, also the concept of risk should be applied (Reisinger et al., The Concept of Risk in the IPCC Sixth Assessment Report: A Summary of Cross-Working Group Discussions. IPCC, 2020). This would increase the value of

this review, would make it more consistent with the broader, comprehensive assessments which also address TPs in several places, and provide consistency in this very complex topic.

2) In order to facilitate the implementation of 1) above, it is suggested that the authors add two additional columns in Table 1 before the columns collecting the references: (i) a column stating the uncertainty and the level of confidence, and (ii) a column stating the risk associated with the realization of this particular "cascade step" (from TP1 to TP2).

**We thank the reviewer for this suggestion that indeed makes our assessment better comparable to established IPCC notions. Therefore, we have applied the following changes to our manuscript.**

**Regarding uncertainty****: We agree that adding levels of confidence to Tab. 1 increases the comparability and current state of research, which we added as a new column to Tab. 1 in the manuscript. It is, however, important to note that uncertainties in tipping points are large, and therefore uncertainties in tipping cascades are even larger. Most of the research on cascading transitions in climate has been performed using conceptual models, and more complex process-based models like EMICs or GCMs only start to become available (see manuscript ll 605-608 and ll 639-657). Therefore, our current assessment needs to remain qualitative using the notion of Fig. 2 of Mastrandrea et al., 2011.**

**Regarding risk****: We also agree that another column (in Tab. 1) stating the risk associated with the realization of a particular "cascade step" would increase the impact of this work. Unfortunately, we feel that adding a credible level of risk is beyond the scope of the current literature. What we can say is the following: while confirmation or rejection through future research is necessary, it cannot be ruled out that interactions between climate tipping elements destabilize the Earth system in addition to climate change effects on individual tipping elements. Therefore, also tipping cascades cannot be ruled out when tipping thresholds of first tipping elements are transgressed through ongoing global warming (we discuss this in ll 639-657). Providing a risk assessment will only be possible (i) for a given level of global warming or a given temperature trajectory, and (ii) with dedicated studies that assess a particular interaction because more complex process-based models like EMICs or GCMs only start to become available (see manuscript ll 605-608). However, we consent that future updates on this review should aim at such risk assessments.**

COMMENT 5: 3) In the present-day example depicted in Fig. 4. the TP 2 "Coastal retreat" is illustrated (more on this below). This is missing from the map of Fig. 1 and the spatio-temporal diagram of Fig. 2. Since this is one of only two examples illustrated in a Figure, it would be important to include this in the two first figures.

4) Whether "coast retreat" is a TP or not is hard to qualify. Currently, the case for a TP is not sufficiently well made or convincing.

**Following this reviewer (as well as other reviewer) recommendation, we moved a strongly shortened version of this section to our new section 2.3.2 (Arctic sea ice → Greenland Ice Sheet and permafrost). We substantiated this section with additional references and added the (highly uncertain) links from the Arctic sea ice to the Greenland Ice Sheet and the permafrost to our Figs. 1-3 and Tab. 1. The reworked text can be found in ll 217-229. The rest of the old section 4 has been placed in the supplementary information due to its more speculative nature.**

COMMENT 6: 5) Line 130: here some basic refs are missing, e.g.: Hu, A.X., et al., Energy balance in a warm world without the ocean conveyor belt and sea ice, Geophys. Res. Lett., 40, 6242-6246, 2013.

**We have added the additional reference to this part of the manuscript in line 150.**

COMMENT 7: 6) line 320: increased freshwater flux and AMOC reduction, along with a southward shift of the ITCZ appears many times in this manuscript. Please reduce redundant repetition.

**We have tried to avoid repetition where possible.**

COMMENT 8: 7) lines 338-339: "relationship" is vague. Please clarify.

**This was indeed unclear in the old version of the manuscript. We have clarified the sentences accordingly in the revised version (see ll 375-377):**

**For example, while the linear effect of ENSO on the Indian Summer Monsoon rainfall has weakened, the effect of ENSO on the West African Monsoon rainfall has increased in recent decades. Both effects need to be further tested in paleoclimate reconstructions,**

COMMENT 9: 8) line 348: The high-latitude response of the hydrological cycle to an increase of GHGs is pretty robust (see IPCC, 2021). You presumably mean the regional response based on a catchment area perspective. Any ref for this?

**While the hydrological cycle in high latitudes is relatively robust (as the reviewer correctly noted), the representation of soil hydrology and its effect on Arctic and subarctic climate is not, as newest literature shows (de Vrese et al., 2023, The Cryosphere). We have clarified this in the new version of the manuscript, see ll 386-388.**

COMMENT 10: 9) line 372-378: this is an uncritical repetition of claims and speculations regarding the "hothouse" as a future possibility, promoted by Steffen et al. in 2018 and since then reiterated in several other papers (e.g. Kemp et al., 2022). The authors should provide the appropriate caveats, or else point to original research – not perspective papers or such – that simulate such effects. Repeating catch words without firm evidence should not feature in a review article and is not useful for the progress in this important topic.

**This is a valid point and it is important to underscore the hypothetical nature of this section. While the section after the mentioned lines already listed the caveats of the hothouse possibility (see ll 421-434), we have now strengthened the hypothetical nature and added some additional references (see ll 413-420). We also added that to our knowledge, no multi-millennial simulation with a model incorporating the elements and feedbacks described above has yet been conducted to test this scenario. This calls for experiments across the model hierarchy (see ll 431-432).**

COMMENT 11: 10) line 378: the breakup of stratocumulus decks, proposed by Schneider et al, is an interesting hypothesis valid in very limited areas in the subtropics. Furthermore, the tipping appears to occur at CO2 levels above 1200 ppm which is very high. Whether or not this purported instability, shown in a limited area LES model only, would translate into a large-scale effect is currently a speculation. This review should present this accordingly.

**The reviewer is correct and we have adapted our sentence on Schneider et al. (2019) accordingly by mentioning the preconditions for their strong temperature feedback: CO2 levels must be around 1200 ppm or larger, see ll 417-420.**

COMMENT 12: 11) Figure 3 illustrates a specific example in the paleoclimate record and panel b) provides some linkages. Some of these linkages are more robust, others are highly uncertain. It would be important to reflect this in, e.g., the line thickness of the arrows. Otherwise, the impression is given that all these links are equally well understood and quantified by, e.g., model simulations.

**The reviewer is right that some of the linkages are better constrained than others. We have replaced arrows by dashed arrows to make clear which linkages are less constrained as compared to others. We have further added a figure on the Eocene-Oligocene transition, where all linkages are very uncertain (therefore all of them are represented by dashed arrows). Please see manuscript figures 4 and 5.**

COMMENT 13: 12) line 614: "We conclude that tipping elements interact …" This appears as a very strong statement of fact. The authors provide no evidence for such robustness and hence a more cautious formulation should be chosen.

**We agree with the reviewer that a cautious notion of our results is necessary. At the same time, we were requested to more clearly lay out consequences of our results to policy makers (Comment 9 of reviewer #1: should policy makers be worried about tipping cascades?). Therefore, we have adapted our manuscript along the following lines (see ll 639-657):**

1. **Tipping elements interact across scales in space and time (see Fig. 1, Tab. 1).**
2. **The majority of tipping element linkages are destabilizing (13 destabilizing, 2 stabilizing, 4 unclear). Therefore, we conclude that tipping elements should not only be studied in isolation, but more emphasis has to be put on potential interactions, in particular given that the majority of links between tipping elements appear to be**

**destabilizing. Thus, tipping cascades cannot be ruled out when tipping thresholds of first tipping elements are transgressed through ongoing global warming. However, decisive for the possibility of a tipping cascade is that a first tipping element shows signs of disintegration, i.e. tipping is initiated. Cryosphere tipping elements would be such candidates (Wunderling et al., 2021, ESD; Wunderling et al., 2023, Nat. Clim. Change).**

3. **Because studies as of now are based on conceptual models, we need detailed assessments of tipping elements and their interactions in process-based models (EMICs or ideally GCMs). This is currently underway, for instance within the so-called TIPMIP endeavor (see ll 605-608).**

COMMENT 14: 13) line 634: "tremendous". Why should a non-quantitative study based on questions and conversations (expert elicitation) be of "tremendous value"? This is overselling this type of information gathering. Of more value would certainly be targeted simulations across a hierarchy of models, careful parameter and sensitivity studies, and large ensemble simulations. The justification of the "tremendous value" is that this would provide "direct expert input". Such "direct expert input" should be reflected in the authorship of, e.g., review papers.

**The reviewer is right that the word *tremendous* is misplaced here. We removed it. We further think that an admixture of different methods is best to reduce uncertainties in tipping cascades. Probably the two best ways are (i) process-based modeling by GCMs or EMICs if models are available (agreeing with the reviewer) or (ii) direct observations or early warning signs of cascading transitions. As long as such methods are not available, also simpler models, ensemble approaches and expert knowledge are important to consider (see ll 658-672).**

COMMENT 15: 14) line 676: This review paper ends by referencing a highly speculative piece which was a perspective paper, provocative and stimulating debate. However, it did not contain quantitative analysis or original research. Therefore, it should be considered an opinion piece and thus be treated in a scientific review accordingly. By citing such work in later scientific journals without the proper qualifiers lends undue support for what originally was a stimulating or provocative idea. Clearly, such work can and should be cited, but the context must be given appropriately.

**We agree with the reviewer and omitted the perspectives/opinion piece by Kemp et al from the last sentence (see line 713).**

---

## Author Response (AR2)

Dear Reviewers, dear Editor,

We are grateful for the positive evaluation of our manuscript and the further reviews. Please find a detailed point-by-point response to the remaining comments of the reviewers below. In addition to this response document we have attached the revised manuscript version, and marked the changes in blue. We thank reviewers for their constructive feedback which has helped in improving this manuscript further.

On behalf of all coauthors,
Nico Wunderling, Anna von der Heydt

**Reviewer #1:**
accept as is
**We thank the reviewer for this positive evaluation of our work and previous suggestions.**

**Reviewer #2:**
The authors have done an excellent job at responding to my comments. I only have two remaining concerns:
**We thank the reviewer for this positive assessment and agree with the two remaining points made. Please find our response below.**

1. The new definition of tipping points (paragraph beginning line 26 in the manuscript with tracked changes) remains poor. The definition does not align with Levermann et al.'s, despite the authors' claim. Levermann et al. write: "Tipping elements are regional-scale features of the climate that could exhibit a threshold behaviour in response to climate change—that is, a small shift in background climate can trigger a large-scale shift towards a qualitatively different state of the system." But the definition in the present manuscript does not include any notion of 'small shift causing a large change'. I suggest the authors update their definition. I also recommend to exclude the term "nonlinear" from the definition (it is certainly not synonymous with "self-amplifying feedback" as the current manuscript implies).
**We thank the reviewer for this important point and regret our definition was not entirely clear yet. We have accordingly adapted our definition so that it now suits the Levermann et al. (2012) definition. We have also removed the term nonlinear to not imply that this would mean the same as *self-amplifying feedback*, see ll 28-32.**

2. The new result that centennial-scale cascades are possible if warming exceeds 2C is important but buried in a very long paragraph in the Discussion. I suggest:
- splitting the paragraph into smaller ones to make the main conclusions more clear -- or, explaining the new result first in Results
**We agree and have restructured the respective discussion paragraph into three main enumerated findings that can be found easily, see ll 646-660.**

- briefly including the new result in the abstract
**We have sharpened our abstract in line with our three main findings from the discussion, see ll 10-15.**

- explaining why this result does not contradict Wang et al. (is it because these cascades may lead to impacts but not large temperature feedbacks?)
**Our results are in line with Wang et al. (2023, Reviews of Geophysics) and their summary in Table 13 because of the following reason: (i) First, tipping cascades cannot be expected within years as is stated in Wang et al.. They can rather not be ruled out on timescales of centuries to millennia for warming levels of 1.5-2.0°C. Only for very high levels of sustained warming (significantly beyond 2.0°C), timescales of tipping cascades**

**on the order of many decades to a few centuries cannot be ruled out. (ii) Second, some Earth system elements may not exhibit tipping behavior but can still be responsible for important Earth system interactions involving tipping elements (such as ENSO→Coral reef/Amazon rainforest interaction or Arctic sea ice→AMOC). In this respect, interactions between (debated tipping) elements can destabilize subsequent climate components. (iii) Third, there are temperature feedbacks that can be expected if tipping events occur (Wunderling et al., 2020; Steffen et al 2018) but they are either limited, taken already into account in complex IPCC-type Earth system models, or act on long timescales (centuries to millennia).**

**Therefore, we believe that our assessment in this review is in line with Wang et al..**

**Lastly, Wang et al. state in Table 13 that the *level of understanding* for tipping cascades is low, which warrants the updated knowledge on tipping element interactions performed in this review.**

**We have added this in a brief statement to our manuscript, see ll 661-662.**

**Reviewer #3:**
accept as is
**We thank the reviewer for this positive evaluation of our work and previous suggestions.**

**Reviewer #4:**
Re-Review: The authors have revised their paper substantially which has led to significant improvements. My final comments primarily concern missing citations that are suggested in order to make this review more comprehensive.
**We thank the reviewer for this positive evaluation of our work and are happy to include the remaining comments by the reviewer into our revised manuscript. In particular, the additional references help to sharpen our manuscript further.**

1.       line 19: cite Stocker & Schmittner 1997. This study is relevant as it showed thresholds associated with both warming levels and speed.
**This is a very relevant study but related to the AMOC specifically.**

2.       line 35: suggest to add the following to the sentence: …. but also due to their potentially dramatic consequences on regional and local scales.
**We have added this sentence including "...including impacts on biosphere and human societies.", see ll 38-39.**

3.       line 42: cite Stocker & Johnsen 2003 that showed the global nature of abrupt changes in the North Atlantic region due to the thermal bipolar seesaw.

**We have added this reference in l 47.**

4.      line 70: suggest to specify this statement by ending the sentence with: … leading to a rapid local response to the slow large-scale changes
**We have added this sentence, see l 75.**

5.      line 91: future projects: specify, e.g. TipMIP, and WhatifMIP
**We agree and have added the respective references in l 97.**

6.      line 114: *group of long-standing experts": are these co-authors of this paper? If so, mention this explicitly. If not, who were they? As it stands this statement lacks transparency
**By this we mean the co-authors of this study. We have rephrased the according sentence, see ll 118-120.**

7.      line 141: Stommel 1961 should be cited as it was the frist to describe the salt advection feedback
**We have added this reference in l 146.**

8.      line 158: the important role of the sub-polar gyre should be mentioned here. cite Born & Stocker 2014 J. Phys Oceanography, Liu & Fedorov 2022
**We have added the role of the Atlantic subpolar gyre and the according references in ll 163-164.**

9.      line 160: …. underestimated, as even moderate warming may push the AMOC across a threshold (Romanou et al., 2023, J. Climate)
**This is indeed an interesting result that even such moderate warming may push the AMOC outside its Holocene state, and possibly towards a tipped state. However, the paper does not discuss the GIS melt effects on the AMOC.**

10.     line 164; 176: cite Stocker & Johnsen, 2003, Pedro et al., 2018
**We have added these references in l 170 and l 181.**

11.     line 180: cite Sutter et al, 2023, Nature Climate Change
**We have added these references in l 185.**

12.     line 249: … with consequences on the carbon inventory (Bozbiyik et al., 2011, Clim. Past)
**We have added the sentence and reference in l 254.**

13.     line 509: what is the measure of "smaller"? The best temperature reconstruction for Greenland during the SDO events is Kindler et al. (2014, Clim. Past) shows several DO equal of larger warming than B/A
**We agree that the measure *smaller* does not account for all B/A events. We therefore omitted it from the sentence, see l 515.**

14.     line 510: reference wrong as these are model simulations. Suggest that the appropriate reference is Barker et al., 2011, Science
**We thank the reviewer for catching this and replaced the reference by Barker et al., 2011, Science, see l 516.**

15.     line 525: should mention that DO could also be self-sustained oscillations, see Vettorietti 2022, Nature Geoscience
**We have added the sentence and reference in ll 531-532.**

16.     line 607: you may cite a recent EMIC development: Pöppelmeier et al., 2023, J. Climate
**Indeed! We have added this reference in l 614.**

17.     line 689: this statement ignores eg RAPID and OSNAP for AMOC monitoring, as well as satellite laser altimetry of ice stream velocities over Greenland and Antarctic ice sheets.
**This is a very valid point that the reviewer raises here. We therefore altered our statement but are convinced of the necessity for further observational efforts in the future to be able to cover all relevant scales in time and space as well as additional observational records for tipping elements. We now write (see ll 703-706):**
**First, data from recent Earth observation efforts (e.g. for AMOC by the RAPID and OSNAP programs (Srokosz et al., 2015), or for ice stream velocities by satellite laser altimetry from ICESat/ICESat-2 (Adbalati et al., 2010; Schutz et al., 2005)) may need to be extended to cover more variables relevant to Earth system tipping elements as well as better covering the relevant temporal and spatial scales.**

18.     line 692-694: here efforts to develop km-scale models need to be mentioned. While not feasible today, with growing computing resources, TP simulations will become feasible within the next decade. Cite Slingo et al, 2023 and Hewitt et al, 2023, both in Nature Climate Change
**We agree and have added the according references together with a statement of these indeed very exciting developments in Earth system science, see ll 710-712.**